# Rapid evolution of A(H5N1) influenza viruses after intercontinental spread to North America

Ahmed Kandeil [1,2,9], Christopher Patton [1,3,9], Jeremy C. Jones [1,9], Trushar Jeevan[1,9], Walter N. Harrington [1,9], Sanja Trifkovic [1], Jon P. Seiler[1], Thomas Fabrizio [1], Karlie Woodard[1], Jasmine C. Turner[1], Jeri-Carol Crumpton[1], Lance Miller[1], Adam Rubrum[1], Jennifer DeBeauchamp[1], Charles J. Russell [1], Elena A. Govorkova [1], Peter Vogel [4], Mia Kim-Torchetti[5], Yohannes Berhane[6,7], David Stallknecht[8], Rebecca Poulson[8], Lisa Kercher [1] & Richard J. Webby [1,3] ✉

Highly pathogenic avian influenza A(H5N1) viruses of clade 2.3.4.4b underwent an explosive geographic expansion in 2021 among wild birds and domestic poultry across Asia, Europe, and Africa. By the end of 2021, 2.3.4.4b viruses were detected in North America, signifying further intercontinental spread. Here we show that the western movement of clade 2.3.4.4b was quickly followed by reassortment with viruses circulating in wild birds in North America, resulting in the acquisition of different combinations of ribonucleoprotein genes. These reassortant A(H5N1) viruses are genotypically and phenotypically diverse, with many causing severe disease with dramatic neurologic involvement in mammals. The proclivity of the current A(H5N1) 2.3.4.4b virus lineage to reassort and target the central nervous system warrants concerted planning to combat the spread and evolution of the virus within the continent and to mitigate the impact of a potential influenza pandemic that could originate from similar A(H5N1) reassortants.

The detection of highly pathogenic A/Goose/Guangdong/1/1996 (GsGD) A(H5N1) influenza viruses in Southern China in 1996 preceded two and a half decades of virus evolution characterized by geographic expansion and contraction and causing significant economic losses and the depopulation of billions of poultry[1]. Over time, viruses of the GsGD H5-lineage have evolved into several transient, phylogenetically distinct hemagglutinin (HA) clades[2]. Genotype turnover driven by virus reassortment has generally occurred only sporadically, but such events occurred with increased frequency among clade 2.3.4.4 yielding A(H5Nx) viruses possessing different neuraminidases (NA), most commonly N1, N6, or N8. Although geographic spread of GsGD H5-lineages is dynamic, clade 2.3.4.4b viruses have consistently expanded since 2020. In the 4 months prior to February 2022, 30 countries or territories across Asia, Europe, and Africa reported the detection of viruses of this clade in birds[3]. Most of these 2.3.4.4b outbreaks had been caused by A(H5N1) viruses, with the noticeable exception of the

[1]Department of Infectious Diseases, St. Jude Children's Research Hospital, Memphis, TN 38105, USA. [2]Center of Scientific Excellence for Influenza Viruses, National Research Centre, Giza 12622, Egypt. [3]Department of Microbiology, Immunology, and Biochemistry, University of Tennessee Health Science Center, Memphis, TN 38105, USA. [4]Comparative Pathology Core, St. Jude Children's Research Hospital, Memphis, TN 38105, USA. [5]National Veterinary Services Laboratories, Animal and Plant Health Inspection Service (APHIS), US Department of Agriculture (USDA), Ames, IA 50011, USA. [6]National Centre for Foreign Animal Disease, Winnipeg, MB R3E 3M4, Canada. [7]Department of Animal Science, University of Manitoba, Winnipeg, MB R3T 2N2, Canada. [8]Southeastern Cooperative Wildlife Disease Study, Department of Population Health, College of Veterinary Medicine, The University of Georgia, Athens, GA 30602, USA. [9]These authors contributed equally: Ahmed Kandeil, Christopher Patton, Jeremy C. Jones, Trushar Jeevan, Walter N. Harrington. ✉e-mail: richard.webby@stjude.org

outbreaks of A(H5N6) in China which have been associated with human infections. Of note, A(H5N6) viruses of clade 2.3.4.4h had been the previously predominant clade in poultry in China and had also caused human infections[4].

In December 2021, A(H5N1) viruses were detected in poultry and a gull in Eastern Canada. The virus responsible was closely related to 2.3.4.4b viruses identified in Europe in the spring of that year[5]. This was only the second time that a GsGD H5-lineage virus was detected in birds in the Americas; the first such occurrence was in 2014, when the infected birds had crossed the Bering Strait and entered the Pacific flyway. After the initial 2021 detection in Newfoundland, infected wild birds were reported in the U.S. south atlantic states[6] and in several other U.S. regions soon thereafter. The purpose of this study was to understand the course of the genetic and phenotypic evolution of the 2.3.4.4b viruses as they spread throughout North America (NAm). We demonstrate that westward intercontinental 2.3.4.4b expansion resulted in reassortment with NAm wild bird viruses yielding reassortants with diverse ribonucleoprotein gene mixtures. The resulting viruses have distinct in vitro phenotypes including increased virus replication rates and pH of activation, but most concerningly, they cause severe disease outcomes with dramatic neurologic involvement in mammalian animal models.

## Results

### Virus pathogenesis and transmission in chickens and ferrets

We first conducted a risk assessment to determine the pathogenic and transmission properties of two early isolates in birds and mammals; A/American wigeon/South Carolina/22-000345-001/2021 (Wigeon/SC/21) and A/bald eagle/Florida/W22-134-OP/2022 (Eagle/FL/22) were collected in December 2021 and February 2022, respectively (Table 1). White leghorn chickens and ferrets were initially used, both of which are susceptible to a variety of influenza viruses and are critical animal models among influenza risk assessment pipelines (IRAT, TIPRA)[7, 8]. Both chicken-to-chicken and chicken-to-ferret transmission were examined. Naïve chickens were placed in the same cage as inoculated birds, and ferrets were housed in a separate cage 12 inches away from inoculated chickens (Supplementary Fig 1A). Both viruses replicated robustly in inoculated chickens but transmitted inefficiently to naïve contact chickens (Supplementary Fig. 1B). Chickens ($n = 3$ per virus) directly inoculated with either Wigeon/SC/21 or Eagle/FL/22 died within 48 h post-inoculation (hpi). Two of twelve contact chickens

paired with Wigeon/SC/21 inoculated chickens shed virus and met euthanasia endpoints at 4 or 7 dpi, while two additional non-shedding contacts died at 4 or 10 dpi. Three of twelve contact chickens paired with Eagle/FL/22 inoculated chickens shed virus and met euthanasia endpoints 3- 4 days post-inoculation (dpi); virus was transiently detected in a fourth bird which survived (Supplementary Fig. 1B). No aerosol transmission was detected from infected chickens to naïve ferrets (Supplementary Fig. 1C), suggesting that the risk of transmission at the avian-mammalian interface remains low. We next assessed ferret-to-ferret contact transmission by placing inoculated ferrets in the same cage as naive ferrets. Neither virus transmitted from inoculated ferrets to naïve direct-contact ferrets (Fig. 1A, and Supplementary Table 4). Surprisingly, there were dramatic differences in disease severity among directly inoculated ferrets between the two viruses. Consistent with a recent report[9], Wigeon/SC/21 infection resulted in mild disease and the ferrets survived (Fig. 1B). Only one of three Wigeon/SC/21 inoculated ferrets exhibited weight loss (Fig. 1C), and upper respiratory tract virus shedding was detected until 5 or 7 ($n = 1$ ferret) dpi (Fig. 1D). In contrast, Eagle/FL/22 inoculation resulted in rapid weight loss (Fig. 1C), lethargy, and severe neurologic symptoms, including ataxia and hindlimb paralysis (Supplementary Table 4). By 7 dpi, all Eagle/FL/22 inoculated ferrets reached humane endpoints (Fig. 1B).

The differential pathogenicity of the two viruses was reflected in the higher mean nasal wash titers at 1, 3, and 5 dpi for Eagle/FL/22 (4.2 to 5.2 $\log_{10}$ TCID$_{50}$/mL) than for Wigeon/SC/21 (1.7 to 3.4 $\log_{10}$ TCID$_{50}$/mL) ($P < 0.0001$) (Fig. 1D). Eagle/FL/22 inoculated ferrets also had higher viral loads in turbinate, tracheal, and lung samples ($P \leq 0.001$), and virus was observed in brain tissues at 3 dpi (mean titer, 6.2 $\log_{10}$ TCID$_{50}$/g) and 5 dpi (mean titer, 8.2 $\log_{10}$ TCID$_{50}$/g) (Fig. 1E). In contrast, Wigeon/SC/21 inoculation led to virus being detected primarily in turbinate and tracheal tissues, trace amounts in the lung at 5 dpi only, and no virus detected in brain tissues (Fig. 1E).

### Genotypic and antigenic analyses of clade 2.3.4.4b viruses

To begin to understand molecular determinants contributing to the higher virulence of Eagle/FL/22, sequence analysis was performed on A(H5N1) wild bird viruses from this outbreak, which included Wigeon/SC/21 and Eagle/FL/22. These data revealed that reassortment event(s) among H5 clade 2.3.4.4b viruses and NAm wild bird influenza viruses had occurred soon after introduction of the A(H5N1) viruses into NAm (Fig. 2). Whereas Wigeon/SC/21 had the same genotype as the first viruses detected in Canada and those detected in Europe in the spring of 2021, Eagle/FL/22 had undergone reassortment, and acquired NAm wild bird-lineage polymerase basic 2 (PB2), polymerase basic 1 (PB1), and nucleoprotein (NP) genes (Fig. 2A).

In total, we identified four different viral genotypes among the 58 A(H5N1) viruses sequenced in this study (Fig. 2B). All viruses maintained the parental Eurasian-origin (EA) HA, NA, matrix (M), and nonstructural (NS) gene segments, but they had different combinations of polymerase and NP gene segments of either EA or NAm origin. In all cases except for PB2, for which two phylogenetically distinct genes were detected (Fig. 2A), the NAm gene segments were monophyletic, suggesting that a single or minimal number of reassortment events had occurred, with the resulting viruses spreading geographically. The NAm-lineage proteins of A(H5N1) viruses had no markers associated with increased virulence in mammalian hosts[10, 11], and all viruses remained antigenically homogeneous, as determined by hemagglutination inhibition (HAI) assays (Supplementary Table 1).

### Pathotyping of additional clade 2.3.4.4b viruses in ferrets

To further assess if virulence was linked to acquisition of NAm gene segments, we used the ferret model to pathotype four additional A(H5N1) viruses of genotypic diversity represented later in the outbreak period (Fig. 3A). These viruses were A/Fancy Chicken/

**Table 1 | North American HPAI A(H5N1) clade 2.3.4.4b viruses used in this study**

| Group[a] | Virus | Genotype[b] | Abbreviation |
|---|---|---|---|
| 1 | A/American wigeon/South Carolina/22-000345-001/2021 | EA | Wigeon/SC/21 |
| | A/Bald eagle/Florida/W22-134-OP/2022 | EA/Nam (PB1, PB2, NP) | Eagle/FL/22 |
| 2 | A/Fancy Chicken/Newfoundland/FAV-0033/2021 | EA | Ck/NL/21 |
| | A/Bald eagle/North Carolina/W22-140/2022 | EA/NAm (PB2, NP) | Eagle/NC/22 |
| | A/Red-shouldered hawk/North Carolina/W22-121/2022 | EA/NAm (PB1, PB2, NP) | Hawk/NC/22 |
| | A/Lesser scaup/Georgia/W22-145E/2022 | EA/NAm (PB1, PB2, PA, NP) | Scaup/GA/22 |

[a]Arbitrary numbering relating to the original ferret inoculation experiments described in Fig. 1 (Group 1), and Fig. 3 (Group 2).
[b]The gene segments acquired from avian influenza viruses from NAm wild birds are indicated in parentheses.
*EA* Eurasian-origin, *NAm* North American-origin, *PB1* polymerase basic 1 gene, *PB2* polymerase basic 2 gene, *PA* polymerase acidic gene, *NP* nucleoprotein gene.

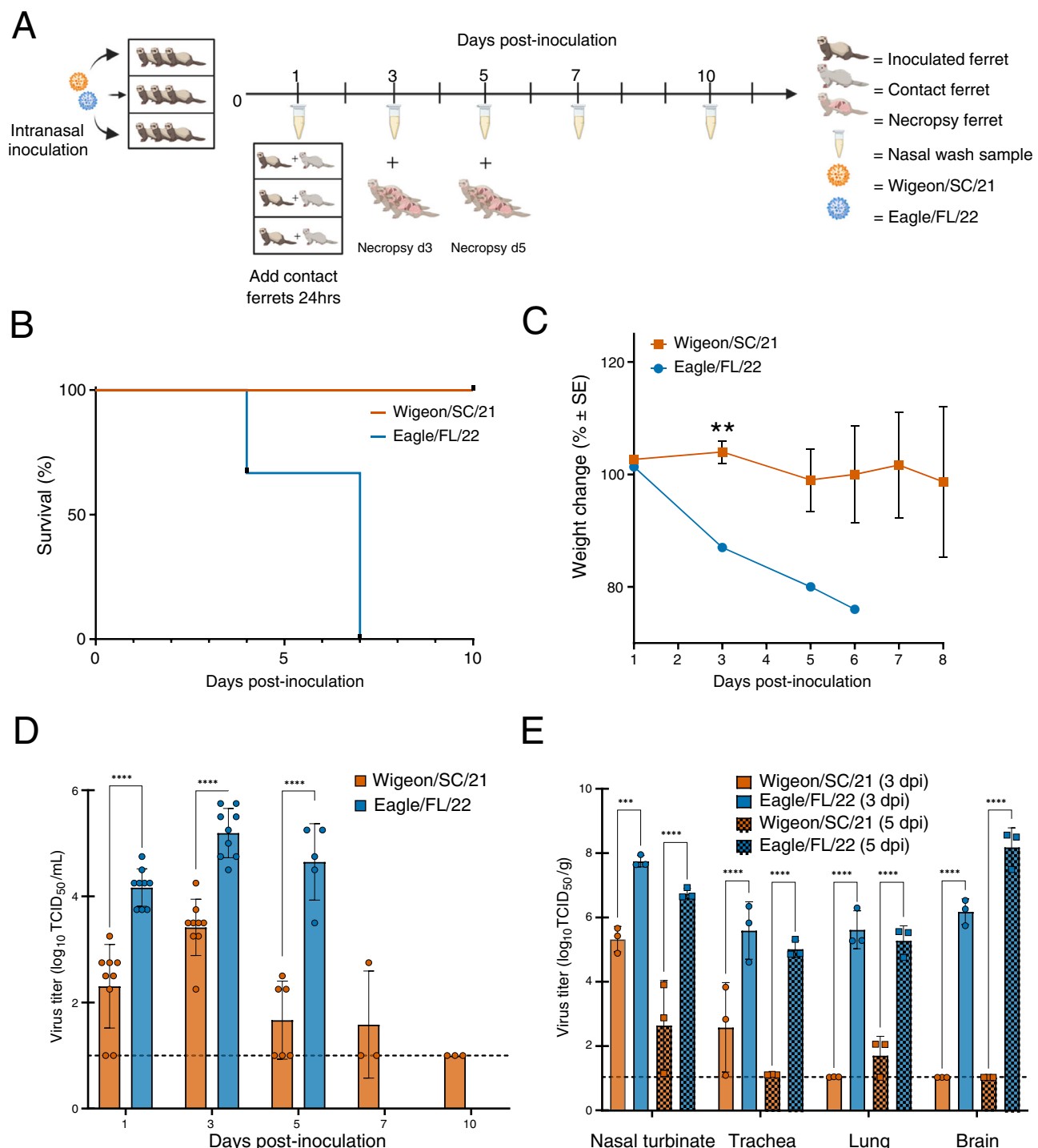

**Fig. 1 | Pathogenicity of North American HPAI Influenza A(H5N1) clade 2.3.4.4b Wigeon/SC/21 and Eagle/FL/22 viruses in ferrets. A** Experimental design of ferret pathogenesis and transmission. At 0 dpi, ferrets ($n = 9$ per virus) were inoculated with $10^6$ EID$_{50}$ units of A(H5N1) virus. Three inoculated ferrets were individually co-housed with 3 naïve contact ferrets beginning 1 dpi. Clinical course of infection was monitored, and nasal wash samples were taken at indicated time points from both inoculated and contact ferrets. The remaining inoculated ferrets were euthanized at 3 dpi and 5 dpi ($n = 3$ per time point per virus) for viral titration in tissues. **B** Survival and **C** weight changes of inoculated ferrets ($n = 3$ per virus). Ferret weights every

≈48 h were used to calculate percentage of weight change from the initial mean weight at 0 dpi. Ferret weight values are the average ± SE for each group. *P* values for weight change were calculated using an unpaired *t*-test. **P < 0.01. **D** Infectious viral titers from nasal washes ($n = 3$–9 ferrets per virus, mean virus titer [log$_{10}$ TCID$_{50}$/mL] ±SD) and **E** infectious viral titers from tissues ($n = 3$ ferrets per virus, mean virus titer [log$_{10}$ TCID$_{50}$ per g of wet tissue]). Symbols represent each individual animal's titer. Dashed lines indicate the lower limit of virus titer detection (1.0 log$_{10}$ TCID$_{50}$/mL). *P* values for viral titers were calculated using two-way ANOVA with Tukey's multiple-comparison post hoc test. ***P < 0.001, ****P < 0.0001.

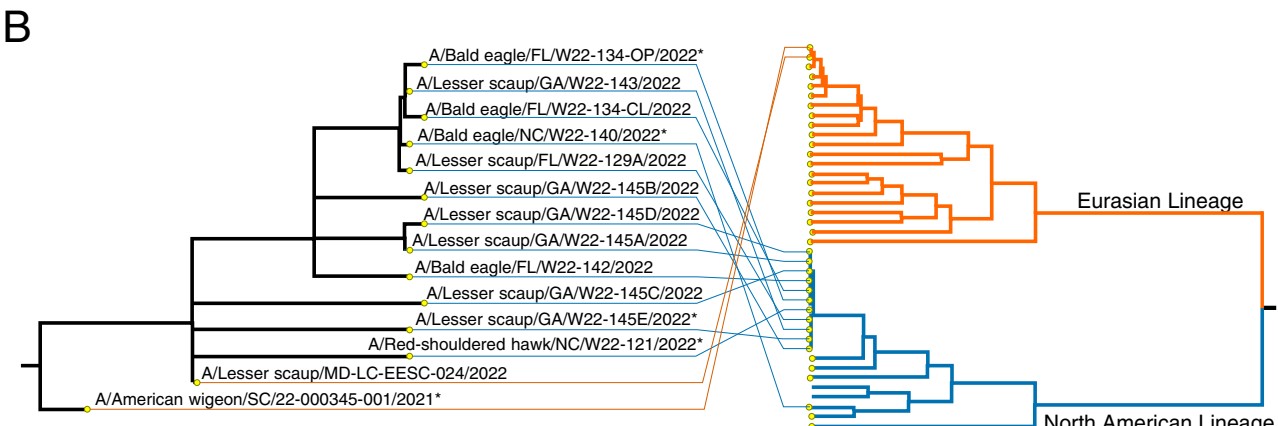

**Fig. 2 | Genotypic diversity among North American HPAI Influenza A(H5N1) clade 2.3.4.4b viruses. A** Genotypic diversity of 58 A(H5N1) viruses (some sequences generated directly from clinical material). A list of viruses used in this study and their GenBank or GISAID accession numbers are provided in Supplementary Tables 3 and 4, respectively. The colors denote common genotypes, with the orange color representing genes of Eurasian lineage and blue representing genes of NAm lineage (the different shades of blue represent phylogenetically distinct NAm genes). **B** Representative tangle plot showing the association of A(H5N1) virus HA genes (left side) with PB2 genes (right side) of Eurasian and NAm lineages. Nodes on the PB2 tree represent A(H5N1) viruses and representatives of other subtypes. Asterisks (*) indicate viruses that were used in subsequent experiments.

Newfoundland/FAV-0033/2021 (Ck/NL/21; native EA constellation; with the same genotype as Wigeon/SC/21); A/Red-shouldered hawk/North Carolina/W22-121/2022 (Hawk/NC/22; with the same genotype as Eagle/FL/22); A/Lesser scaup/Georgia/W22-145E/2022 (Scaup/GA/22), with PB2, PB1, polymerase acidic (PA), and NP of NAm wild bird-lineage; and A/Bald eagle/North Carolina/W22-140/2022 (Eagle/NC/22), with PB2 and NP of NAm wild bird-lineage (Table 1). In general, the number of NAm gene segments acquired was strongly associated with disease severity in ferrets including degree of weight loss and the number of animals that ultimately succumbed to disease (Fig. 3B, C), with the least number of clinical signs and mortality observed with Ck/NL/21, which has no NAm genes, and the most significant signs and high mortality observed with Scaup/GA/22, which possesses four NAm genes (Fig. 3B, C, Supplementary Figs. 2, 3, and Supplementary Table 4). This trend extended to viral replication in tissues. Peak mean nasal wash titers were 2.9, 3.8, 5.4, and 6.0 $\log_{10}$ TCID$_{50}$/mL for Ck/NL/21, Eagle/NC/22, Hawk/NC/22, and Scaup/GA/22 viruses, respectively (Fig. 3D). All EA/NAm viruses were detected in multiple organs confirming systemic viral spread, while the native constellation virus Ck/NL/21 had very low or trace amounts of virus outside the respiratory tract (Fig. 3E). Additionally, infection with the EA/NAm viruses resulted in higher temperature increases than Ck/NL/21 along with respiratory and neurologic symptoms (Supplementary Table 4).

**Pathology**
Virulence trends were also reflected in the histopathologic examination of the upper respiratory tract (URT), lower respiratory tract (LRT), and extrapulmonary tissues of ferrets infected with viruses of different genotypes. The most virulent virus, Scaup/GA/22, produced severe

necrotizing lesions throughout the URT (Supplementary Fig. 3A, B) and LRT (Supplementary Fig. 4A, B) that correlated with robust viral antigen staining. Histopathology was less severe in ferrets infected with Hawk/NC/22, which produced fewer necrotizing lesions, limited multifocal clusters of neuroepithelium in URT (Supplementary Fig. 3C, D) and well demarcated LRT lesions (Supplementary Fig. 4C, D), corresponding to less extensive viral antigen staining. Abundant antigen staining was observed throughout the central nervous system (CNS) of ferrets inoculated with Scaup/GA/22 (Supplementary Fig. 5A, B), whereas antigen staining for Hawk/NC/22 was limited to the olfactory bulb and olfactory cortex of inoculated ferrets (Supplementary Fig. 5C, D). Eagle/NC/22 produced even less severe histopathology, with only a few small foci of virus-positive cells being observed in the URT olfactory neuroepithelium and no lesions or virus antigen staining extending into the LRT (Supplementary Fig. 3E, F, and Supplementary Fig. 4E, F). Finally, in ferrets inoculated with Ck/NL/21, both the URT and LRT respiratory epithelia were devoid of lesions or cells staining for viral antigen, except for a single small focus of virus antigen-positive cells in the olfactory neuroepithelium (Supplementary Fig. 3G, H, and Supplementary Fig. 4G, H).

**Virus Pathogenesis in the mouse model**
To investigate whether the pathogenicity and virulence in ferrets was mirrored in other influenza mammalian models, we inoculated BALB/c mice with each virus. Viruses that caused 100% lethality in ferrets and those that had acquired increasing numbers of NAm gene segments, including Scaup/GA/22, Eagle/FL/22, and Hawk/NC/22, had the lowest 50% lethal doses (LD$_{50}$s) in mice (i.e., less virus was required for lethality) at 3.8, 2.2, and 4.6 $\log_{10}$ EID$_{50}$/mL, respectively. Additionally,

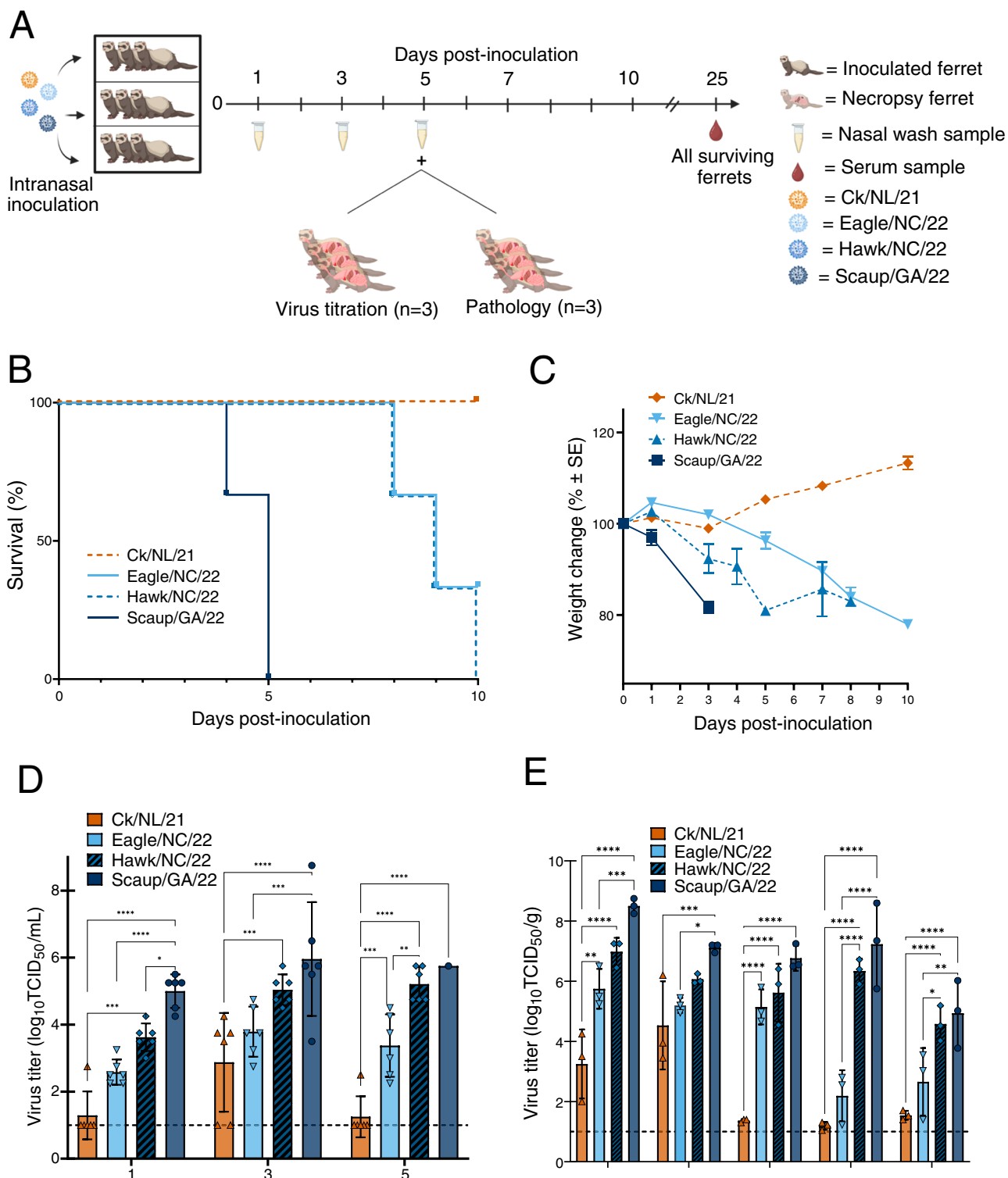

**Fig. 3 | Impact of different detected genotypes of North American HPAI Influenza A(H5N1) clade 2.3.4.4b viruses on pathogenicity in ferrets.** **A** Experimental design of ferret pathogenesis and transmission. At 0 dpi, ferrets ($n = 9$ per virus) were inoculated with $10^6$ EID$_{50}$ units of A(H5N1) virus. Clinical course of infection was monitored, and nasal wash samples were taken at the indicated time points. Ferrets ($n = 3$ per virus per analysis) were euthanized at 5 dpi for viral titration in tissues and pathology (Supplementary Figs. 3–5). **B** Survival and **C** weight changes of inoculated ferrets ($n = 3$ per virus). Ferret weights every ≈48 h were used to calculate percentage of weight change from the

initial mean weight at 0 dpi. Ferret weight values are the average ± SE for each group. **D** Infectious viral titers from nasal washes ($n = 6$ ferrets per virus, except for Scaup/GA/22 at 5 dpi, for which $n = 1$, mean virus titer [log$_{10}$ TCID$_{50}$/mL] ±SD) and **E** infectious viral titers from tissues ($n = 3$ ferrets per virus per time point, mean virus titer [log$_{10}$ TCID$_{50}$ per g of wet tissue]). Symbols represent each individual animal's titer. Dashed lines indicate the lower limit of virus titer detection (1.0 log$_{10}$ TCID$_{50}$/mL). $P$ values were calculated using two-way ANOVA with Tukey's multiple-comparison post hoc test. $^*P < 0.05$, $^{**}P < 0.01$, $^{***}P < 0.001$, $^{****}P < 0.0001$.

these viruses induced neurologic symptoms in mice, along with weight loss and virus replication in the lungs and brains. Viruses causing moderate or no lethality in ferrets, including Eagle/NC/2, Ck/NL/21, and Wigeon/SC/21, had the highest mouse $LD_{50}$s at >6 $\log_{10}$ $EID_{50}$/mL (Supplementary Figs. 6 and 7).

## Phenotypic properties of clade 2.3.4.4b viruses

Several influenza virus characteristics are known to contribute towards mammalian infection and spread, including receptor binding properties, pH of HA activation, and polymerase activity[12–17]. Wigeon/SC/21 and Eagle/FL/22 bound strongly to sialic acid receptors with a 3'SLN linkage to the underlying sugar (preferred by avian viruses), but poorly to 6' SLN-linked sialic acid receptors (preferred by human viruses) (Fig. 4A, B). The pH of HA activation and inactivation segregates with host adaptation[17], and the pH of HA activation for A(H5N1) viruses in avian species is 5.6–6.0[12]. Consistent with this, both Wigeon/SC/21 and Eagle/FL/22 had a relatively high pH of HA activation of 5.8 by syncytium assay (Fig. 4C and Supplementary Fig. 7). These values were slightly higher than the pH of HA activation of the control human virus A/California/04/2009 (CA/04) (H1N1)pdm09 (pH 5.6) and substantially higher than the pH of HA activation of most human-adapted influenza viruses[14]. In contrast, Wigeon/SC/21 and Eagle/FL/22 had virus inactivation pH values of 4.4, substantially lower than that of the CA/04 (H1N1)pdm09 virus (pH 5.4) (Fig. 4C). A virus inactivation pH that is lower than the HA activation pH is rare but has been observed with some swine H1 and H3 isolates and a bat A(H9N2) isolate[18]. The polymerase activities of Wigeon/SC/21 and Eagle/FL/22 were not statistically different when measured in the reporter gene assay with transiently expressed proteins (Fig. 4D). However, we detected differences in the replication rates of whole virus in non-differentiated Calu-3 cells (Fig. 4E) and primary differentiated human airway cultures (Fig. 4F). At 48 hpi, cell cultures inoculated with viruses that had the most NAm gene segments (Scaup/GA/22, Eagle/FL/22, and Hawk/NC/22) had the highest viral loads, followed by those inoculated with Eagle/NC/22, and then those inoculated with viruses containing no NAm gene segments (Ck/NL/21 and Wigeon/SC/21). Additionally, at 72 hpi, differentiated human airway cultures infected with Scaup/GA/22 and Eagle/FL/22 exhibited higher viral loads as compared to a representative seasonally circulating rg-A/Texas/71/2017 (H3N2) virus (Fig. 4F) ($P \le 0.001$). In contrast, Widgeon/SC/21 titers trended lower, while CK/NL/21 titers were statistically lower than rg-A/Texas/71/2017 (H3N2) ($P = 0.001$).

Protective antibody immunity to influenza virus targets the HA and NA and is an important consideration when assessing the risk posed by zoonotic influenza viruses[16]. Using a set of 48 human serum samples obtained from blood donors aged 18 to 46 years, we examined the presence of such cross-reactive antibodies. As expected, there were generally no significant neutralizing titers (serum dilution >1:20) to H5 HA protein (Fig. 4G). In contrast, an enzyme-linked lectin assay (ELLA) targeting antibodies to the N1 protein revealed the geometric mean titers (GMTs) of antibodies against both Wigeon/SC/21 and Eagle/FL/22 to be equivalent to those of antibodies against the seasonal CA/04 (H1N1)pdm09 NA in this serum panel (Fig. 4H). The NA proteins of the A(H5N1) viruses of clade 2.3.4.4b and those of A(H1N1)pdm09 viruses have 89.6% amino acid identity, with considerable conservation at some antigenic sites[19, 20].

## Antiviral susceptibility of clade 2.3.4.4b viruses

To evaluate whether the available antiviral therapies would be effective against the A(H5N1) clade 2.3.4.4b viruses, we examined the phenotypic susceptibility of these viruses to antivirals from two U.S. FDA–approved classes: the NA inhibitors (NAIs) oseltamivir and zanamivir and the active metabolite of the PA endonuclease inhibitor baloxavir marboxil. All six viruses tested (both reassortant and native constellations) had $IC_{50}$ or $EC_{50}$ values comparable to those of drug-susceptible human A(H1N1)pdm09 influenza reference viruses

(Table 2). Additionally, genotypic M gene analysis revealed the absence of amino acid changes associated with reduced susceptibility to the adamantane class of drugs (Table 2).

## Discussion

With the spread of A(H5N1) viruses throughout the United States and the detection of an infection in a human[21], the increased virulence of the reassortant viruses is of considerable concern. Except for pockets of endemic clade activity in South and Southeast Asia, clade 2.3.4.4b viruses have predominated over other A(H5Nx) clades over the past 18 months. Clade 2.3.4.4b viruses have become entrenched in Asia, in Europe, and probably in parts of Africa. The WHO update (covering September 2021 to February 2022) reported 26 cases of 2.3.4.4b infection in humans, comprising 25 cases of A(H5N6) infection in China and one case of A(H5N1) in the United Kingdom, demonstrating the zoonotic transmission potential of these viruses[3]. From a public health perspective, the increased pathogenicity of the reassortant A(H5N1) viruses is of significant concern. However, this is tempered by the avian virus-like characteristics of the viruses with respect to their receptor binding preference and their pH of HA activation. Modification of these characteristics are likely required to enable sustained human-to-human transmission, although only a few amino acid changes among various influenza proteins are needed to switch these properties during adaptation in mammals[22, 23]. In addition to the representative viruses we assayed phenotypically, we did not identify any mutations previously associated with 6'SLN-linked sialic acid binding or alterations in pH of HA activation in the viruses we sequenced. To date, a single case of human A(H5N1) infection in an individual involved in poultry culling has been detected in North America during the current outbreak[21]. This individual underwent isolation and oseltamivir treatment and recovered after experiencing only minor symptoms. Publicly deposited sequence from samples collected from this case (albeit partial and missing PB2 and NP segments) were of Eurasian origin. Zoonotic transmission is likely to continue if this virus remains present in North American wild birds, and vigilance must be maintained as these birds continue their migrations. While direct translation of animal model findings to humans must be done cautiously, future zoonotic infections with the reassortant A(H5Nx) viruses may well include severe cases. Consistent with this possibility is the recent identification of a severe reassortant virus infection of a child in Ecuador[24].

The A(H5Nx) clade 2.3.4.4 viruses originally identified in NAm in 2014–2015 ultimately disappeared from avian species for reasons that are unclear[25], but previous experience is unlikely to predict future events in this situation. Like the current NAm clade 2.3.4.4b viruses, the 2014–2015 viruses reassorted soon after detection on the continent, but this reassortment was not associated with changes in mammalian pathogenicity[26]. In contrast, our study with contemporary 2.3.4.4b viruses revealed that reassortants containing increasing numbers of NAm gene segments exhibited enhanced virulence with neurological involvement in mammalian models, including ferrets. However, these viruses retained avian-like receptor specificity, did not possess molecular markers of mammalian adaptation, or the ability to transmit between ferrets. Clade 2.3.4.4b viruses have thus far been associated with infections within a European mink farm[27], spread among aquatic mammals in the Americas[28–30], and reported identification from a variety of other NAm mammals including foxes, skunks, bobcats, mountain lions, and bears[31]. The increasing prevalence of 2.3.4.4b viruses are also changing the dynamics of disease in Europe, with the potential for transition from epizootic to enzootic status[32]. Our data highlight how quickly things can change in a natural system, and the potential for further A(H5Nx) reassortment and phenotypic diversification will only increase as the unprecedented global distribution of these viruses broadens.

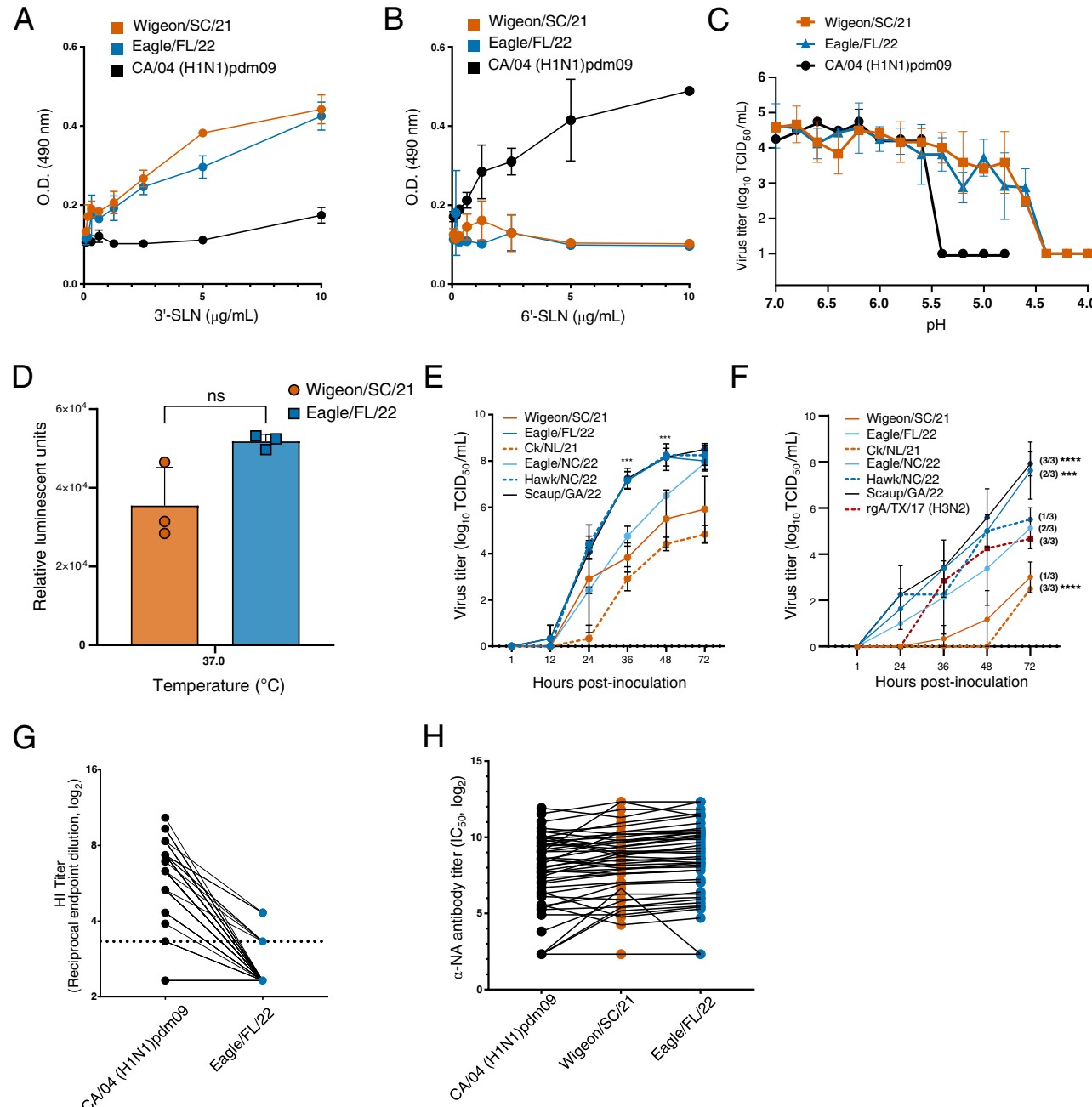

**Fig. 4 | Phenotypic properties of North American HPAI Influenza A(H5N1) clade 2.3.4.4b viruses. A, B** Solid-phase binding of A(H5N1) viruses to biotinylated sialylglycopolymers **A** 3′-SialLacNAc-PAA-biotin (3′-SLN) or **B** 6′-SialLacNAc-PAA-biotin (6′-SLN), representing galactose-linked sialic acids α2,3-SA (the avian virus preferred receptor) and α2,6-SA (the human virus preferred receptor), respectively. The data are shown as the mean ± SD from duplicate wells and representing one of two independent experiments. **C** Kinetics of pH inactivation of the Wigeon/SC/21, Eagle/FL/22, and CA/04 (H1N1)pdm09 viruses at 37 °C. The data are shown as the mean ± SD from triplicate wells representing one of three independent experiments. **D** Minireplicon polymerase activities of Wigeon/SC/21 and Eagle/FL/22 at 37 °C. The data are shown as the mean ± SD of 3–4 measurements over the hypothesized pH range for avian viruses, and 2 measurements over previously described ranges for control virus CA/04 (H1N1)pdm09, and representing one of three independent experiments. NS = not significant as determined by paired, two-tailed

*t*-test. **E** Viral replication kinetics in Calu-3 cells. Cells were inoculated at an MOI of 0.001 and incubated at 37 °C. The data are shown as the mean ± SD from triplicate wells and representing one of two independent experiments. **F** Viral replication kinetics in primary differentiated human airway cultures. Cultures were inoculated at an MOI of 0.005 and incubated at 37 °C. The data are shown as the mean ± SD from triplicate culture inserts and representing one of two independent experiments. The number of inserts with viral replication out of the total is indicated in parentheses. Statistical significance (one-way ANOVA) was determined by comparison to rg-A/Texas/71/2017 (H3N2) at 72hpi. **G, H** Neutralizing antibody levels in human serum samples (*n* = 48) against HA protein (as measured by HI assay, dotted line indicates limit of detection of 1:10 serum dilution) or NA protein (as measured by ELLA assay). Points joined by lines represent values for the same individual for the individual antigens tested. *P < 0.05, ***P < 0.001, ****P < 0.0001.

## Methods

### Ethics statements and animal husbandry
All animal studies including co-housing of chickens and ferrets in the same bioisolator were approved by the St. Jude Children's Research Hospital Institutional Animal Care and Use Committee (IACUC, protocol number 428) in accordance with the guidelines established by the Institute of Laboratory Animal Resources, approved by the Governing Board of the US National Research Council, and carried out by

**Table 2 | Susceptibility of North American HPAI A(H5N1) clade 2.3.4.4b to approved antiviral drugs**

| Influenza A virus | Subtype | Susceptibility to antiviral drugs | | | | | | M2 inhibitor |
|---|---|---|---|---|---|---|---|---|
| | | PA inhibitor (baloxavir)[a] | | NA inhibitors[c] | | | | (amantadine)[e] |
| | | Mean EC$_{50}$ ± SD (nM) | Fold change[b] | Oseltamivir | | Zanamivir | | |
| | | | | Mean IC$_{50}$ ± SD (nM) | Fold change[d] | Mean IC$_{50}$ ± SD (nM) | Fold change[d] | |
| Wigeon/SC/21 | H5N1 | 0.16 ± 0.05 | <1.0 | 1.25 ± 0.65 | 4.4 | 0.22 ± 0.03 | 1.1 | S |
| Eagle/FL/22 | H5N1 | 0.22 ± 0.06 | 1.1 | 0.98 ± 0.57 | 3.4 | 0.19 ± 0.04 | 1 | S |
| Ck/NL/21 | H5N1 | 0.09 ± 0.09 | <1.0 | 1.26 ± 0.62 | 4.4 | 0.20 ± 0.04 | 1 | S |
| Eagle/NC/22 | H5N1 | 0.52 ± 0.04 | 2.7 | 1.07 ± 0.56 | 3.8 | 0.18 ± 0.05 | <1.0 | S |
| Hawk/NC/22 | H5N1 | 0.27 ± 0.03 | 1.4 | 1.34 ± 0.68 | 4.7 | 0.20 ± 0.04 | 1 | S |
| Scaup/GA/22 | H5N1 | 0.17 ± 0.08 | <1.0 | 1.13 ± 0.90 | 4 | 0.21 ± 0.04 | 1.1 | S |
| *Reference influenza viruses* | | | | | | | | |
| rg-A/CA/04 (PA I38-WT) | (H1N1) pdm09 | 0.19 ± 0.04 | 1 | N/A | N/A | N/A | N/A | R (S31N) |
| rg-A/CA/04 (PA I38T) | (H1N1) pdm09 | 15.32 ± 3.13 | 80.6 | N/A | N/A | N/A | N/A | R (S31N) |
| A/Denmark/524/2009 (NA H275-WT) | (H1N1) pdm09 | N/A | N/A | 0.29 ± 0.65 | 1 | 0.20 ± 0.02 | 1 | R (S31N) |
| A/Denmark/528/2009 (NA H275Y) | (H1N1) pdm09 | N/A | N/A | 118.81 ± 0.65 | 416.4 | 0.24 ± 0.05 | 1.2 | R (S31N) |

[a]Reduction in plaque formation number in virus-infected MDCK cells at 72 hpi or 96 hpi. Mean values (nM) ± SD from three independent dose-response curves are presented

[b]Fold change relative to the baloxavir susceptibility of reverse genetics (rg)-derived A/CA/04 (H1N1)pdm09 virus containing PA I38 (a baloxavir-susceptible genotype). A/CA/04 (H1N1)pdm09 containing PA I38T (a genotype with reduced susceptibility to baloxavir) is provided for comparison.

[c]Reduction in fluorescence signal from NA-cleaved MUNANA substrate. Mean values (nM) ± SD from four independent dose-response curves are presented.

[d]Fold change relative to the NAI susceptibility of A/Denmark/524/2009 (H1N1)pdm09 containing NA H275 (a genotype with normal inhibition by NAIs). A/Denmark/528/2009 (H1N1)pdm09 containing NA H275Y (a genotype for which inhibition by oseltamivir is highly reduced) is provided for comparison.

[e]Susceptibility to adamantanes was based on the absence of substitutions at M2 residues known to mediate adamantane resistance (amino acids 26, 27, 30, 31, and/or 34).

*N/A* not applicable to specific inhibitor analysis, *S* amantadine-susceptible virus, *R* amantadine-resistant virus. Amantadine resistance-associated mutations are shown in parentheses.

trained personnel working in a United States Department of Agriculture (USDA)-inspected Animal Biosafety Level 3+ animal facility in accordance with all regulations established by the Division of Agricultural Select Agents and Toxins (DASAT) at the USDA Animal and Plant Health Inspection Service (APHIS), as governed by the United States Federal Select Agent Program (FSAP) regulations (7 CFR Part 331, 9 CFR Part 121.3, 42 CFR Part 73.3). Animal holding rooms were on a 12 h light/12 h dark cycle, with dry-bulb temperatures set below each species lower critical temperature to minimize potential heat stress (71 °C ± 1 °C with alarm points beyond this value). Room humidity was set to 45%, with active humidification of each room allowing ≥40% humidity during winter months. Human sera were purchased from a commercial provider (BioIVT, Hicksville, NY) who played no role in our study. The specimens were not collected specifically for our study, and we had no access to the subject identifiers linked to the specimens. This study utilized USDA-classified select agents and A(H5N1) viruses used herein are subject to the guidelines of, and compliance with, requirements discussed in Title 9 (CFR Parts 121 [Possession, Use, and Transfer of Select Agent Toxins] and 122 [Importation and Transportation of Controlled Organisms and Vectors]).

## Cell culture
Madin−Darby canine kidney (MDCK) cells (ATCC CCL-34) and African green monkey kidney (Vero) cells (ATCC CCL-81) were grown in culture in Modified Eagle's Medium (MEM) (CellGro) supplemented with 5% fetal bovine serum (FBS) (HyClone), 1 mM L-glutamine, and 1× penicillin/streptomycin/amphotericin B (Gibco). Human embryonic kidney (HEK293T) cells (ATCC CRL-3519) were grown in culture in OptiMEM (Gibco) supplemented with 10% FBS and 1× penicillin/streptomycin/amphotericin B. Human airway epithelial (Calu-3) cells (ATCC HTB-55) were grown in culture in MEM supplemented with 10% FBS, 1 mM L-glutamine, 1 mM sodium pyruvate, and 1× penicillin/streptomycin/amphotericin B. All cells were maintained at 37 °C in 5%

$CO_2$. Primary differentiated human airway cultures were purchased from MatTek and maintained as discussed in the replication kinetics section.

## Influenza virus propagation and titration
The A(H5N1) influenza viruses were isolated from wild birds in the allantoic cavities of 10-day-old embryonated chicken eggs (eggs) at 35 °C for up to 48 h. Seasonal influenza A(H1N1) or rg-A(H3N2) viruses were grown in MDCK cells at 37 °C for up to 72 h. Viral titers were determined by injecting 0.1 mL of 10-fold dilutions of virus into the allantoic cavities of 10-day-old eggs and then calculating the 50% egg infectious dose (EID$_{50}$) or by inoculating MDCK monolayers and then calculating the 50% tissue culture infectious dose (TCID$_{50}$) by the method of Reed and Muench[33]. The lower limit of virus detection was 1.0 log$_{10}$ TCID$_{50}$/mL or 1.0 log$_{10}$ EID$_{50}$/mL.

## Replication kinetics
Calu-3 cells (approximately $9 \times 10^5$ cells/well in 6-well plates) were washed with phosphate-buffered saline (PBS) and inoculated with virus (MOI of 0.001) in 2.0 mL of infection medium (MEM, 1% bovine serum albumin [BSA, Sigma-Aldrich], 0.3–1 μg/mL TPCK-trypsin [for non-A(H5N1) viruses], 1× penicillin/streptomycin/amphotericin B). One hour later, the inocula were removed, the monolayers were washed with PBS, and the supernatants were replaced with 3.0 mL of infection medium. Primary differentiated human airway cultures (MatTek, AIR-100) on trans-well inserts were cultured at an air-liquid interface using manufacturer provided medium in the basal chambers and no medium the apical chambers. Cells were washed and inoculated (MOI of 0.005), with the addition of a low pH 2.0 physiological saline wash to inactivate residual virus. No medium was added back to the apical chambers. At each timepoint, 200 μL of BEBM (Lonza) supplemented with 1% BSA was added to the apical chambers for 15 min at 37 °C, then harvested. All cells or insert supernatants were sampled at 1 to 72 hpi as indicated,

and virus titers ($log_{10}$ $TCID_{50}$/mL) were determined in MDCK cells by the method of Reed and Muench[33]. Data are presented for one of two independent experiments conducted using triplicate wells/inserts for each time point and for each virus among each cell line.

### Receptor binding assay

Flat-bottom, 96-well immune assay plates (ThermoFisher) were coated with 10 μg/mL fetuin (Sigma-Aldrich) at 4 °C overnight then washed three times with washing buffer (PBS with 0.01% Tween 80). Plates were blocked with PBS containing 1% BSA then incubated at 4 °C overnight with 32 HA units of the Wigeon/SC/21 A/(H5N1), Eagle/FL/22 A/(H5N1), or CA/04 A/(H1N1)pdm09 viruses. Plates were incubated with biotinylated sialylglycopolymers: 3′-SialLacNAc-PAA-biotin (3′-SLN) (Glycotech) or 6′-SialLacNAc-PAA-biotin (6′-SLN) (Glycotech), serially diluted in the reaction buffer (PBS with 0.02% Tween 80, 0.02% BSA, and 5 μM oseltamivir carboxylate). After incubation for 2 h at 4 °C, the plates were washed and incubated for 1 h with horseradish peroxidase-conjugated streptavidin (Invitrogen; diluted 1:2000) in blocking solution. After washing, the plates were incubated with o-phenylenediamine dihydrochloride (Sigma-Aldrich) at room temperature for 10 min. The reaction was stopped by adding 1 N $H_2SO_4$ (Fisher Chemicals), and the absorbance was measured at 490 nm in a Synergy H1 microplate reader (BioTek).

### Syncytium assay

Syncytium formation in vero cells was assayed as described[34]. Briefly, vero cells were seeded in a 12-well plate ($1.5 \times 10^5$ cells/well) and incubated at 37 °C overnight. The cells were then washed with PBS, inoculated with each virus at an MOI of 3.0, and incubated at 37 °C for 1 h. The inocula were then removed and replaced with 1.0 mL of infection medium. At 6 hpi [for the A(H5N1) viruses] or 24 hpi [for the CA/04 (H1N1)pdm09 virus], the supernatants were removed, the cells were washed with PBS, 1.0 mL of infection medium supplemented with 5 μg/mL of TPCK-treated trypsin was added, and the plates were incubated at 37 °C for 10 min. The infection medium was then removed, and residual trypsin was neutralized with 1.0 mL of MEM supplemented with 5% FBS. The supernatants were replaced with 0.5 mL of PBS, the monolayers were examined visually to verify their confluence, then the PBS was replaced with an equal volume of pH-adjusted buffer (pH 5.5–6.0) for 5 min. The pH buffer supernatants were then removed, and the cells were washed with PBS and incubated in 1.0 mL of MEM supplemented with 5% FBS at 37 °C for 3 h. The cells were fixed and stained using the Hema 3 Stat Pack (Fisher Scientific). Micrographs were obtained using a Nikon Eclipse TS100 microscope with a Zeiss Axiocam ERc 5 s camera at 100× total magnification. The HA activation pH was defined as the highest pH at which syncytium formation was observed. The resolution of this assay was 0.1 pH units. Mock-inoculated cells (treated with PBS only) and cells inoculated with CA/04 (H1N1)pdm09 virus were used as negative and positive controls, respectively.

### pH of inactivation

The A(H5N1) influenza viruses were standardized to a concentration of $1.6 \times 10^7$ $TCID_{50}$/mL and incubated with pH-adjusted buffer at 37 °C for 1 h. Samples were neutralized by adding infection medium. The infectious titers of neutralized samples were determined in MDCK cells by $TCID_{50}$ assay[33]. The resolution of this assay was 0.2 pH units. The CA/04 (H1N1)pdm09 virus was used as a positive control.

### Enzyme-linked lectin assay (ELLA)

The presence of NA-specific antibodies was determined in ELLAs[35]. Tested human sera samples were purchased from a commercial vendor (BioIVT). The vendor had no role in the study, the samples were not collected specifically for this study, and no subject identifiers linked to the specimens were available to the investigators. Briefly, flat-

bottom, 96-well plates (Thermo Scientific) were coated with fetuin (Sigma-Aldrich) at 25 μg/mL in 0.1 M PBS at 4 °C for 48 h. After blocking, heat-inactivated sera (inactivated at 56 °C for 1 h) were serially diluted in Dulbecco PBS (DPBS) (Gibco) supplemented with 1% BSA and 0.5% Tween 20 and added to the plates. This was followed by the addition of a standardized reverse genetics (rg)–derived H6Nx antigen (virus). The plates were incubated at 37 °C for 16–18 h and washed with PBS containing 0.05% Tween 20, then horseradish peroxidase-conjugated peanut agglutinin (Sigma-Aldrich) was added at 1 μg/mL and the plates were incubated for 2 h at room temperature. The plates were then washed and 3,3′,5,5′-tetramethylbenzidine (Sigma-Aldrich) was added. The color reaction was stopped after 10 min by adding 1 N $H_2SO_4$. The plates were read at 450 nm for 0.1 s in a Synergy H1 microplate reader. The NI titers were defined as the reciprocal of the last dilution that resulted in at least 50% inhibition.

### Phenotypic susceptibility to neuraminidase inhibitors

The neuraminidase (NA) inhibitors oseltamivir carboxylate (oseltamivir) and zanamivir were purchased from MedCh Express. NA susceptibility was determined by fluorometric assay using the substrate 2′-(4-methylumberlliferyl)-α-D-N-acetylneuraminic acid (MUNANA) (Sigma-Aldrich) with additional modifications[36, 37]. Influenza viruses were standardized to equivalent NA activity and incubated with 10-fold dilutions of each NA inhibitor (at concentrations of 5 pM to 50 μM). The fluorescence signal of the released 4-methylumbelliferone was measured using a Synergy H1 microplate reader at excitation/emission (Ex/Em) wavelengths of 360 nM/460 nM. The 50% inhibitory concentrations ($IC_{50}$s) were estimated from dose-response curves by using the sigmoidal, four-parameter logistic non-linear regression equation (in GraphPad Prism v9). The results are representative of the combined values of at least three independent dose-response curves and include influenza A/Denmark/524/2009 (H1N1)pdm09 (NA H275, NAI susceptible), and A/Denmark/528/2009 (H1N1)pdm09 (NA H275Y, NAI reduced susceptibility) as reference viruses.

### Phenotypic susceptibility to endonuclease inhibitors

The active metabolite of baloxavir marboxil, baloxavir acid (BXA), was purchased from MedChem Express. BXA susceptibility was determined by plaque-reduction assay. MDCK cells ($10^6$ cells/well in 6-well plates) were inoculated with a virus volume previously determined to yield approximately 50–100 plaque-forming units (PFUs). At 1 hpi, the cells were washed and overlaid with MEM containing 0.45% immunodiffusion-grade agarose (MP Biomedical), 0.1% BSA, 1 μg/mL TPCK-trypsin (for non-A(H5N1) viruses), and 10-fold dilutions of BXA (1 pM to 1 μM). At 72 hpi, the overlays were removed, and the cell monolayers were stained and fixed with 1% crystal violet, 10% formaldehyde. The number of PFU per well was calculated, and the 50% effective concentrations ($EC_{50}$s) were determined by using the log (inhibitor) versus response logistic nonlinear regression equation (in GraphPad Prism v9). The results are representative of three independent dose-response curves and include rg–derived CA/04 A(H1N1) pdm09 viruses with the wild-type PA I38 (BXA sensitive) or the PA I38T substitution (reduced susceptibility to BXA) as reference viruses[38, 39].

### Genotypic susceptibility to M2 protein inhibitors

Susceptibility to M2 protein inhibitors (amantadine, rimantadine) was determined from the M2 splice product of the full-length M gene, with analysis of the M2 amino acid substitutions (residues 26, 27, 30, 31, and 34) mediating resistance.

### Generation of reverse-genetics (rg) viruses

Influenza virus gene segments were amplified from viral RNA using gene-specific primers (Integrated DNA Technologies, Supplementary Table 5) and cloned into the influenza A virus rg dual-promoter expression vector pHW2000[40]. To generate rg viruses, plasmids

encoding cDNAs of all eight genomic RNA segments were transfected into HEK293T cells by using Lipofectamine 3000 reagent (Thermo Fisher). At 48 h post-transfection, the cell supernatant was harvested, and 0.2 mL of the supernatant was injected into 10-day-old embryonated chicken eggs to propagate the virus. Viruses used in ELLAs were generated with the HA gene of A/Teal/Hong Kong/w312/1997 (H6N1), all internal gene segments were from A/Puerto Rico/8/1934 (H1N1), and the NA gene was from either Wigeon/SC/21 (H5N1) or Eagle/FL/22 (H5N1). rg-A/H1N1 and rg-A/H3N2 viruses containing PA substitutions conferring baloxavir reduced susceptibility were generated in previous studies[38, 39].

## Minireplicon assay
Influenza virus genes encoding PB1, PB2, PA, and NP were cloned into pHW2000 as described previously, propagated in Top 10 competent cells (Invitrogen), and purified with a HiSpeed Plasmid Maxi Kit (Qiagen). HEK293T cells ($1.5 \times 10^5$ cells/well in a 48-well plate) were transfected (using Mirus TransIT-LT1 transfection reagent) with the virus plasmids, a pPolI-358 NP firefly luciferase reporter gene for influenza polymerase activity (kindly provided by Megan Shaw, Mount Sinai School of Medicine, New York, NY), and a pCMV-β-galactosidase plasmid for transfection control normalization. After 24 hpi at 37°C, the monolayers were lysed with 0.2 mL of passive lysis buffer (Promega) and the supernatants were clarified by centrifugation. The ratio of luciferase activity (as measured with the Luciferase Assay System [Promega]) to β-galactosidase activity was determined using 15 μL of clarified supernatant. Data shown are representative of one of three independent assays using at least triplicate measures for each sample.

## Sanger sequencing
Viral RNA was extracted using a RNeasy Mini Kit (Qiagen), and cDNA was synthesized using a OneStep RT-RCR Kit (Qiagen). PCR amplification of influenza A virus gene segments was performed using primers (sequences available upon request). PCR products were extracted from 1.5% agarose gel by using a QIAquick Gel Extraction Kit (Qiagen). Sanger sequencing was performed by the Hartwell Center at St. Jude Children's Research Hospital on an ABI Prism capillary sequencer (Applied Biosystems) and assembled with DNASTAR Lasergene version 15.3.0, 422.

## Illumina sequencing
Viral RNA was extracted using a RNeasy Mini Kit (Qiagen), and cDNA was synthesized using the Superscript IV First-Strand Synthesis System (Invitrogen). The influenza A virus gene segments were amplified using modified universal primers in a multi-segment PCR as described[41]. PCR products were purified using Agencourt AMPure XP beads according to the manufacturer's protocol (BeckmanCoulter). Libraries were prepared using the Nextera XT DNA Library Prep Kit (Illumina) according to the manufacturer's protocol and sequenced using a MiSeq Reagent Kit v2 (300 cycles) on a MiSeq System (Illumina). Sequencing reads were then quality trimmed and assembled using CLC Genomics Workbench (version 22.0.1). A total of 423 gene segment sequences obtained from 53 isolates in this study were deposited in the Influenza Research Database and are available under GenBank accession numbers presented in Supplementary Table 3.

## Phylogenetic analyses
Sequences other than those found in this study were retrieved from the National Center for Biotechnology Information Influenza Virus Sequence Database and from the EpiFlu database of the Global Initiative on Sharing All Influenza Data (GISAID[42]). Sequences were then aligned, and ends were trimmed to equal lengths with BioEdit sequence alignment editor software (v.7.2.5). Similar sequences, including sequences obtained in this study, were removed, and

phylogenetic relationships were inferred by the neighbor-joining method from 500 bootstrap values; topology was confirmed by the maximum likelihood method[43]; and evolutionary analyses were conducted with the MEGA 7 software[44].

## Tanglegram
Additional North American and Eurasian influenza HA and PB2 gene sequences isolated from May 2021 to May 2022 were obtained from the GISAID database (Supplementary Table 4). Redundant PB2 sequences were eliminated by setting a sequence identity threshold of 97%, using CD-HIT (v.4.8.1)[45]. Twenty-five non-redundant PB2 sequences from the GISAID database were combined with 14 studied North American A(H5N1) sequences by using BioEdit (v.7.2.5). Using the default parameters of Augur (v.15.0.1)[46], the sequences were aligned, and a phylogenetic tree was constructed. Furthermore, a PB2 gene time-resolved tree was generated using TreeTime (v.0.8.6)[47], which is included in Augur. Using exclusively the 14 studied A(H5N1) sequences, an HA gene time-resolved tree was generated through the same Augur process. These data were visualized using Auspice (v.2.29.1)[48].

## Virulence and transmission in ferrets
Four-to-six-month-old outbred influenza-seronegative male ferrets (Triple F Farms, Sayre, PA, USA) were lightly anesthetized with isoflurane and inoculated intranasally with $10^6$ EID$_{50}$ units of A(H5N1) virus diluted in 1.0 mL of PBS. Ferrets ($n = 3$ to 6 per group) were monitored daily for clinical signs of infection. Characteristics monitored included body temperature, weight loss, relative inactivity indices[49], ataxia, respiratory symptoms, stool consistency, and neuropathologic signs. Animals reaching the humane endpoint, according to an IACUC-approved clinical scoring system, were euthanized. Nasal washes were collected from all surviving ferrets at 1, 3, 5, 7, and 10 days post-infection (dpi). Ketamine was used to induce sneezing. At 3 and 5 dpi (in the initial phenotyping experiment) and 5 dpi (in the follow-up experiment), ferrets ($n = 3$ per group) were euthanized, and tissue samples were collected from the respiratory tract (nasal cavity, trachea, and lungs), the central nervous system (a combination of the brain stem, cerebellum, and frontal lobes), and the small intestine (not examined for Wigeon/SC/21 and Eagle/FL/22). Viral titers were determined in MDCK cells by TCID$_{50}$ assay. For transmission studies in the initial phenotyping experiment, three ferrets were inoculated intranasally with A(H5N1) virus as described above and housed individually in separate cages. Direct-contact transmission was assessed by placing a naïve ferret into the cage of the inoculated ferret at 1 dpi. Ferrets were monitored daily, and nasal washes were collected at 1, 3, 5, 7 and 10 dpi, starting just before the inoculated and contact ferrets were co-housed. At 25 days post contact, sera were collected from all surviving inoculated and contact ferrets and treated with receptor-destroying enzyme (Denko) as per manufacturer instructions. Degree of seroconversion was determined by hemagglutination inhibition (HAI) assay[50], and discussed in detail in the Serological Testing section.

## Pathogenicity in mice
Groups of 6-to-8-week-old female BALB/c mice (Jackson Laboratory, Bar Harbor, ME, USA) were lightly anesthetized with isoflurane and inoculated intranasally with 10-fold serial dilutions of virus suspension containing $10^2$ to $10^6$ EID$_{50}$ units of A(H5N1) virus in 30 μL of PBS. After virus inoculation, mice were weighed daily and monitored for mortality (death or loss of ≥25% of their body weight) and any clinical signs of infection for 14 dpi. The 50% mouse lethal dose (LD$_{50}$) was calculated after the 14-day observation period. For viral replication studies, BALB/c mice ($n = 5$ per group) were lightly anesthetized with isoflurane and inoculated intranasally with $10^4$ EID$_{50}$ units of each tested virus in 30 μL of PBS. At 5 dpi, mice were euthanized, and their lungs and brains were collected and homogenized in 1 mL of infection medium. Viral

titers in harvested organs were determined by $TCID_{50}$ assay in MDCK cells.

## Transmission in chickens

Six-week-old specific-pathogen-free (SPF) white leghorn chickens (Charles River Laboratories, CT, USA) ($n = 3$ chickens per virus) were inoculated with $10^6$ $EID_{50}$ units of Wigeon/SC/21 (H5N1) or Eagle/FL/22 (H5N1) virus by the natural routes (0.2 mL intranasally, 0.1 mL intraocularly, 0.1 mL intraesophageally, and 0.1 mL intratracheally). At 1 hpi, naïve chickens ($n = 12$ per virus) were co-housed with the donor chicken and all birds were monitored daily for disease signs. Oropharyngeal and cloacal specimens were collected at 2, 4, 6 dpi, and the infectious viral titers were determined by $EID_{50}$ assay. To evaluate aerosol transmission from chickens to ferrets, naïve ferrets ($n = 3$) were housed in a ferret cage that was placed adjacent to a donor chicken cage (distance between cages was 12 inches) within the same bioisolator. Ferret nasal washes were collected at 2, 4, 6, 8, and 10 dpi, and infectious viral titers were determined by $EID_{50}$ assay. All animals were monitored twice daily for disease signs. At 21 dpi, sera were collected from all ferrets to assess the degree of seroconversion by HAI assay as described below.

## Serologic testing

Sera samples ferrets or chickens were treated with receptor-destroying enzyme II (Denka Seiken Co.) at 37 °C overnight, heat-inactivated at 56 °C for 45 min. Hemagglutination inhibition (HI) titer was determined by incubating 2-fold serial dilutions of serum sample with 25 µl of 4 hemagglutinating (HA) units (HAU) in 96-well U-bottom plates (Corning). Sera and virus mixtures were incubated at room temperature for 45 min prior to addition of a 0.5% solution of chicken red blood cells (Rockland Immunochemicals) in PBS and subsequent incubation at room temperature for 30 min. The HI titers were recorded as the reciprocal of the highest serum dilution where there was complete inhibition of hemagglutination and reported as endpoint doubling dilution or $Log_2$ as indicated in each figure or table.

## Pathology

Ferret lungs and nasal mucosa were fixed via intratracheal/intranasal infusion with 10% neutral-buffered formalin (NBF) followed by continued immersion in 10% NBF. Tissues were routinely processed and embedded in paraffin, sectioned, and stained with hematoxylin and eosin. Serial sections were subjected to antigen retrieval for 30 min at 98 °C before undergoing immunohistochemical labeling of viral antigen, using a primary goat polyclonal antibody (US Biological, Swampscott, MA) against influenza A/USSR/1977 (H1N1) virus at a dilution of 1:1000 and a secondary biotinylated donkey anti-goat antibody (Santa Cruz Biotechnology) at a dilution of 1:200.

## Statistical analysis

Data were analyzed using unpaired $t$-tests, two-way ANOVA with Tukey's multiple-comparison post hoc test, and univariant log-rank analysis (survival curves) in GraphPad Prism v9. Replicates, group comparisons, and $P$ values are listed in each figure legend.

## Reporting summary

Further information on research design is available in the Nature Portfolio Reporting Summary linked to this article.

## Data availability

Data generated in this study are provided in the main manuscript, supplementary information, and/or source data files. GenBank and GISAID sequence accession numbers are provided in supplemental Table 2 and 3. Source data are provided with this paper.

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

## Acknowledgements

We thank Keith A. Laycock for the valuable editing of the manuscript, the staff of the Animal Resources Center for their excellent care of the research animals, and the Hartwell Center for Bioinformatics and Biotechnology at St. Jude Children's Research Hospital for their help with the next-generation sequencing. Graphical Figs. 1, 3, and Supplemental Fig. 1 were created using BioRender.com. This project has been funded in whole or in part with federal funds from the National Institute of Allergy and Infectious Diseases, National Institutes of Health, Department of Health and Human Services, under contracts HHSN272201400006C (R.W.) and 75N93021C00016 (R.W., D.S.), U.S. National Science Foundation (1911955, 2200310, R.W.) award, NIAID/NIH R01AI150745 (R.W.) grant, and by St. Jude Children's Research Hospital and ALSAC (R.W.).

## Author contributions

All authors actively participated in scientific discussion of the manuscript and provided comments, critiques, and/or approvals prior to submissions and revisions. R.W., L.K., R.P., D.S., and P.V. conceptualized the research project(s). A.K., C.P., J.J., T.J., R.W. conceived the methodologies. P.V., L.K., and W.H. provided data visualization. L.K., J.B., E.G., and R.W. administered the projects. A.K., C.P., J.J., T.J., W.H., S.T., J.P., T.F., K.W., J.T., J.C., L.M., A.R., J.D., L.K. acquired the data. R.W., L.K., E.G. wrote the original draft. R.W., L.K., E.G., J.J., C.P., M.T., P.V., W.H. reviewed and edited subsequent drafts. Funding was acquired by R.W. and D.S.

## Competing interests

The authors declare no competing interests.
