## [Peer Review File · Nature Communications]

Reviewer comments, first round review

Reviewer #1 (Remarks to the Author):

Genetic diversification associated with increased mammalian neurovirulence is described in Northamerican HPAIV H5N1 viruses of GsGd clade 2.3.4.4b. A very rich spectrum of molecular functional assays has been employed to describe phenotypic characteristics of natural reassortant viruses with local Northamerican AIV. Neurovirulence and systemic replication capacities in experimentally infected ferrets and Balb/c mice positively correlated with increasing numbers of reassorted polymerase and NP gene segments of Northamerican origin. However, transmissibility between mammals and from chickens to mammals seemed to be limited likely due to avian-adapted receptor binding and pH-related HA activation properties. This, the authors concluded, should have a tempering effect on zoonotic properties of these reassorted viruses. Nevertheless, should the viruses gain mammalian-adaptive mutations, the human population would not have HA H5 cross reactive humoral immunity although widely effective antibody cross reactivity was measured against NA N1. The authors justly point out the velocity and unpredictability of evolution of these viruses once a different geographic compartment has been entered.

The study provides timely and original data based on a very wide spectrum of state-of-the-art molecular and functional assays. The manuscript is very clear and the figures are highly informative. Very few minor aspects might be considered before proceeding to fast publication:

30 – Instead of 2022, shouldn't it read something like "by the end of 2021"?

32 – Here and further in the main body of text, could the authors state whether the reassorted segments stem from wild bird or poultry viruses?

54 – There are also human infections in the same region with A(H5N6) viruses of clade 2.3.4.4h. A short note would help to clarify matters.

262 – The Northamerican poultry population also seems hit hard by these viruses. Would the authors agree that the interface between humans and poultry is more meaningful for exposure and transmission to humans than that with wild birds?

Reviewer #2 (Remarks to the Author):

Review of the manuscript "Rapid evolution of A(H5N1) influenza viruses after intercontinental spread to North America" by Dr Webby and colleagues. We are facing a global spread of avian influenza viruses in wild and domestic birds on an unprecedented scale. This manuscript describes the genetic and phenotypic properties of the H5N1 2.3.4.4b viruses as they spread throughout North America in 2021/2022. The virus acquired new polymerase gene cassettes resulting in different genotypes/phenotypes and exhibiting a wide-range of illness severity in mammal models, from asymptomatic disease to severe neurological disorders and death. In the initial phenotyping experiment, two isolates, representing pure Eurasian (Wigeon/SC/21 from 2021) and reassortant Eurasian/North American (EA/NAm) viruses (Eagle/FL/22 from 2022) (group 1) were selected to infect chickens (n= 3) and ferrets (n=3) and to assess virus transmission. Both viruses were excreted from oropharyngeal and cloacal swabs of primarily inoculated chickens and few sentinel chickens excreted viruses. No ferret-to-ferret or chicken-to-ferret transmission was observed. In ferrets, Wigeon/SC/21 was low virulent, while Eagle/FL/22 was highly virulent and replicated at higher levels in different organs. Although all viruses were antigenically related, sequence analysis of Eagle/FL/22 and newly sequenced viruses (> 50) indicated four different genotypes due to the reassortment of clade EA 2.3.4.4b with polymerase complex segments of local NAm wild-bird viruses. Compared to an EA H5N1, three H5N1 reassortant genotypes were highly virulent and replicated at high levels in ferrets. No significant differences were observed in receptor-affinity/specificity, HA pH-fusion activation, polymerase activity (despite different constellations?), neutralisation with human NA antibodies or sensitivity to oseltamivir, zanamivir and marboxil. The current study provides novel and valuable information on the pathogenesis of different NAm reassortant H5N1 genotypes in ferrets, mice and chickens. However, the study is qualitative. The

number of inoculated chickens (n=3) and ferrets (n=3) is very low and doesn't support robust statistical analysis. The pathogenicity and transmissibility of one of the two viruses (Wigeon/SC/21) in the initial experiment was already tested in ferrets and human cells in a previous study (Ref.#7). Similarly, there are many studies which described the spread and evolution of the recent clade 2.3.4.4b (for example: PMID: 35302933, 36261139, 35821511, 36105667) in North America in 2021/2022 and recombination with the local AIV. However, Potential biological factors for variable virulence were explored, although reproducibility of the results (few replicates) is inadequate and the interpretation of these results is not clear. There were no positive controls in many experiments (e.g. polymerase activity, replication kinetics, pathogenesis/transmission studies in ferrets) or negative control ferrets to accurately assess the changes in bodyweight, temperature and activity. Few sentences are contradictory between the figure legends and the text.

Major comments:

1. While the findings of the pathogenesis studies are highly useful and relatively novel for the recent American H5N1-2022, however, the preliminary biological characterisation did not explain variations in the pathogenicity of different viruses. How to explain these striking variations in the pathogenesis of different reassortants? How to link it to L261-262 "Zoonotic transmission is likely to continue"?
2. Replication of the viruses in primary human cells or organoids should be done. Replication of these highly virulent HPAIV in Calu3 is not enough. The replication kinetics lack positive controls.
3. The experimental design (in chickens and ferrets) is extremely difficult to understand. The total number of animals in each experiment is not clear, particularly for ferrets (Fig 1B). The design should be clearly described. The legends should be clear enough to understand the figures. If possible, a diagram should be provided to explain the design of the experiments. Variation in the inoculation route of chickens Wigeon/SC/21 or Eagle/FL/22 should be justified. This can be the reason for variable pathogenesis of these two H5N1 viruses in chickens. Supplementary Fig S1, naïve birds were added 1 hpi or 1 dpi? 4 naïve chickens were added per donor chickens (legend of Fig S1) or per virus (L415 methods file)?
4. Reproducibility of the results is not rigorous enough. Triplicate wells, triplicate experiments ... duplicates... are vague descriptions. How many independent biological replicates have been done for each assay?
5. The interpretation of the results is very broad and should be rephrased and concentrated.

Minor comments:

Main Text file:

1. Values in the figures (e.g. Figs 1B, 1C, 3B, 3C, supplementary Fig 2, supplementary Fig 7) should be provided as individual titres not as columns with mean+/-SD.
2. L68-73, L410-421: The experimental design is very confusing. It is not clear how many birds were infected? whether all inoculated and sentinel chickens died or survived? What is the mean death time in chickens? Was there any difference in the pathogenicity in chickens (e.g. mean death time after inoculation with different route ..intranasal vs. intraoesophageal?)."
3. L80: "Fig 1B" why 5 to 9 animals are shown?
4. L85: Wigeon/SC/21 (4.3 log₁₀TCID₅₀/mL) is not reflected in Fig 1B
5. L88: "TCID₅₀/mL" is not accurate... /gm, please check the legends of other (suppl.) figures.
6. L119: ref.#8 is not suitable for this sentence.
7. L121: Fig 2: for readability, please indicate the viruses used in the subsequent animal experiments by asterisks or arrows.
8. L146-147: this sentence is not accurate. Fig. 3C shows low CK-NL-21 titres in the brain and intestine.
9. L188: "the pH of HA activation" why the difference of 6 hpi and 24 hpi for the H5N1 and A(H1N1)pdm09 virus? Please explain!
10. L189: ref.#11 is not suitable for this sentence

Supplementary file:

11. A list of viruses (not only the H5N1 mentioned in Table S1) used in this study and their accession numbers should be provided as a supplementary Table.
12. Supplementary Figure S1: the figure and legend are confusing. L34, which values? The numbers refer to the contact animals (4 not 12?) excreted viruses, or seropositive? What about

numbers for primarily inoculated animals? For clarity and transparency, the individual titres should be shown, not only the average +/- SD. "intranasally by natural routes" should be clarified.

13. Supplementary Fig S6: the color coding is very difficult to follow particularly in panels E and F. It is recommended to use open and closed circles, triangles, ... for clarity.

14. Supplementary L141: typo, the title is not complete. The HI test is not described in the materials and methods section. How the test was done and what about the SD?

15. Supplementary Table 3. Please describe how the Relative inactivity index was calculated and whether this index is statistically significance or not. Seroconversion data should be added to this table.

16. Supplementary L205-213: the ATCC numbers should be added.

17. Supplementary L225: MOI 0.01 or MOI 0.001? Please check the legend of figure 4.

18. Supplementary L230-231: The results for the two replicates should be shown.

19. L391: where is the result of the "small intestine" for the initial animal experiment?

20. L395: "every 2 days" is not accurate (10 dpi)

21. Animal welfare issues, L418-419: please describe the housing and distance between chickens (prey) and ferrets (predator) (see PMID: 26202239) and describe the behaviour of animals in this experimental setting.

Reviewer #3 (Remarks to the Author):

Kandeil et al., have performed in-depth characterization of representative H5N1 GsGd clade 2.3.4.4 viruses that have been introduced into North America and spread rapidly in domestic and wild birds. Initial studies are performed with two early isolates, and these studies are expanded to encompass four additional strains. Importantly, via reassortment, some of these viruses have acquired internal genes from avian influenza viruses already circulating in North America. The authors demonstrate that as additional North American lineage internal genes (PB2, PB1, PA, NP) are introduced into these viruses, there was an associated increase in virulence in mammals. The virulence is striking as these viruses are lethal to ferrets, and the HA, NA, M, and NS genes are largely unchanged. Thus, the changes in virulence are most likely due to the internal genes. However, no known mammalian adaptation markers are present in these genes. The authors proceed to perform several analyses that are considered fundamental for gauging pandemic risk. These include changes in antigenicity, receptor-binding preference, mouse lethal dose 50 studies, anti-viral screening, assessment of pH of fusion, and viral polymerase activity. The studies provide a comprehensive overview of these viruses and indicate on-going risk assessment is warranted as the viruses continue to evolve.

Major Comments: None.

Minor Comments:

1) This reviewer would encourage the authors to move some of the figures/tables in the Supplemental Materials to the main manuscript. Several figures and tables in the Supplemental Materials are reference extensively. It would enhance interpretation of the findings if some of this data was in the main manuscript.

2) Lines 74-76. The authors indicate that ferret-ferret and chicken-ferret transmission studies have been performed; however, the data is not present in the manuscript. This text should either be clarified or removed, or the data should be added to the manuscript.

3) Lines 143-144. The manuscript uses several isolates often with similar names. In this section, it would enhance clarity and ease of interpretation, if the peak viral titers for the isolates were presented in the same order as in Figure 3B and C.

4) Line 192. This line appears to indicate that all 6 isolates in the paper were evaluated for receptor-binding preference. However, only 2 H5N1 isolates were used to assess receptor-binding preference. Please revise to clarify.

5) The authors have not mentioned if there are any mutations in the HA that are known to be associated with binding to 2, 6 linked sialic acids and/or that are known to modify the pH of fusion. This should be added to the manuscript.

6) Line 202-203. The authors indicate that the polymerase activity of the viruses were similar over all temperature ranges. This is true when comparing the isolates, but it appears that polymerase

activity at 37 and 40°C are lower than at 34°C. This finding requires more explanation. This observation of reduced activity at higher temperatures is unexpected as birds have higher core body temperatures (38-40°C). Moreover, for most influenza viruses when polymerase assays are performed, the overall level of polymerase activity is usually higher at 37°C relative to 33°C. When mutations such as PB2 E627K are introduced into avian viruses, these viruses usually have elevated activity at 33°C relative to the precursor strains; however, this activity typically is equal or less than that at 37°C. Thus, the authors are encouraged to discuss and/or explain the significance of these findings.

7) Figure 4. It would be very interesting to evaluate viral growth kinetics at 33 or 34°C. This is not required for publication, but would potentially complement the polymerase activity data.

8) Line 230, please indicate "data not shown" when discussing neutralizing titers against HA.

RESPONSES TO REVIEWERS' COMMENTS

Nature Communications manuscript NCOMMS-22-42769-T

Rapid evolution of A(H5N1) influenza viruses after intercontinental spread to North America

We appreciate the helpful comments and suggestions from each reviewer. We have addressed each critique below and incorporated the changes within the manuscript file. Additional details on methods have been included as requested, including graphical representations of experimental setups for three figures (Fig. 1A, Fig. 3A, and Supplementary Fig. 1A). We also conducted additional experiments or added additional data to address reviewer critiques that include: a third cell type (primary human airway cultures grown at air-liquid interfaces) to model replication kinetics, a seasonal virus control in the human airway-culture experiments, and more chicken/ferret titer data within the supplemental materials. Overall, we believe these revisions strengthen the manuscript quality.

REVIEWER COMMENTS

Reviewer #1 (Remarks to the Author):

Genetic diversification associated with increased mammalian neurovirulence is described in North American HPAIV H5N1 viruses of GsGd clade 2.3.4.4b. A very rich spectrum of molecular functional assays has been employed to describe phenotypic characteristics of natural reassortant viruses with local North American AIV. Neurovirulence and systemic replication capacities in experimentally infected ferrets and Balb/c mice positively correlated with increasing numbers of reassorted polymerase and NP gene segments of North American origin. However, transmissibility between mammals and from chickens to mammals seemed to be limited likely due to avian-adapted receptor binding and pH-related HA activation properties. This, the authors concluded, should have a tempering effect on zoonotic properties of these reassorted viruses. Nevertheless, should the viruses gain mammalian-adaptive mutations, the human population would not have HA H5 cross reactive humoral immunity although widely effective antibody cross reactivity was measured against NA N1. The authors justly point out the velocity and unpredictability of evolution of these viruses once a different geographic compartment has been entered.

The study provides timely and original data based on a very wide spectrum of state-of-the-art molecular and functional assays. The manuscript is very clear and the figures are highly informative. Very few minor aspects might be considered before proceeding to fast publication:

RESPONSE: The authors thank Reviewer 1 for succinctly summarizing our study and for the encouraging remarks.

COMMENT #1. 30 – Instead of 2022, shouldn't it read something like “by the end of 2021”?

RESPONSE: We thank the reviewer for pointing out this mistake. We have revised the statement to read: “*By the end of 2021, viruses of this clade were detected in North America, signifying further intercontinental spread*”. (lines 30, 31)

COMMENT #2. 32 – Here and further in the main body of text, could the authors state whether the reassorted segments stem from wild bird or poultry viruses?

RESPONSE: Phylogenetically, the North American gene segments were most similar to viruses from wild birds. We have modified as, “*Here, we show that the western movement of clade 2.3.4.4b was quickly followed by reassortment with viruses circulating in wild birds in North America, resulting in the acquisition of different combinations of ribonucleoprotein genes.*” (lines 31-33). We have also added the “wild bird” qualifier to statements on lines 111, 114, 117, 157, & 158.

COMMENT #3. 54 – There are also human infections in the same region with A(H5N6) viruses of clade 2.3.4.4h. A short note would help to clarify matters.

RESPONSE: As suggested, to provide some clarity we have now added, “*Of note, A(H5N6) viruses of clade 2.3.4.4h had been the previously predominant clade in poultry in China and had also caused human infections⁴.*” (lines 53, 54).

COMMENT #4. 262 – The North American poultry population also seems hit hard by these viruses. Would the authors agree that the interface between humans and poultry is more meaningful for exposure and transmission to humans than that with wild birds?

RESPONSE: This is a very interesting question and one that we have discussed at length. Earlier in the outbreak poultry did appear to be the most likely interface with humans. As time has progressed and the virus has spread very widely in wild birds, it could be argued that substantial risk also exists at the wild bird/animal-human interface. This interface could include hunter killed waterfowl- it is possible many A(H5Nx) virus-contaminated duck carcasses sit in freezers across the US- and scavenger birds including owls, eagles, and falcons which are often rescued if appearing sick. Infection and subsequent severe disease of scavenger mammals also provides an additional exposure risk. We are somewhat hesitant to identify poultry as the major human risk interface, although the numbers of chickens and turkeys that have been exposed and infected is tragic and the sole U.S. human detection arose via contact with chickens. Due to word limits we have not added additional text to the manuscript, but we are certainly prepared to if the Reviewer or Editor prefers.

Reviewer #2 (Remarks to the Author):

Major comments:

COMMENT #1. While the findings of the pathogenesis studies are highly useful and relatively novel for the recent American H5N1-2022, however, the preliminary biological characterisation did not explain variations in the pathogenicity of different viruses. How to explain these striking variations in the pathogenesis of different reassortants? How to link it to L261-262 “Zoonotic transmission is likely to continue”?

RESPONSE: Our original driver for this study was a thorough biologic characterization of these viruses as they entered a new geographic space. We were very surprised by the striking variations in the observed phenotypes. Presently, we do not have an explanation for this variation. Many of the additional assays we presented, for example polymerase and some replication assays, were conducted for this very purpose. Neither suggested a definitive answer, though viruses with more robust replication rates in vitro (i.e. EA/FL/22, Scaup/GA/22) were associated with the most severe morbidity and mortality in ferrets. As detailed in our data,

increased virulence was associated with higher viral load in brain and lung tissue, both likely contributors to the increased disease. The reassortant viruses do not have molecular markers that have been associated with increased zoonotic potential. But with increasing numbers of infected poultry and wildlife, the chances of human exposure to the virus similarly increases. The reviewer makes a good point that we did not link the increased virulence to human zoonotic infection. In response to reviewer comment, we have modified the following discussion to link our findings to human infections (lines 284-291): “*This individual underwent isolation and oseltamivir treatment and recovered after experiencing only minor symptoms. Publicly deposited sequence from samples collected from this case (albeit partial and missing PB2 and NP segments) were of Eurasian origin. Zoonotic transmission is likely to continue if this virus remains present in North American wild birds, and vigilance must be maintained as these birds continue migrations. While direct translation of animal model findings to humans must be done cautiously, future zoonotic infections with the reassortant A(H5Nx) viruses may well include severe cases. Consistent with this possibility is the recent identification of a severe reassortant virus infection of a child in Ecuador²²*”.

COMMENT # 2. Replication of the viruses in primary human cells or organoids should be done. Replication of these highly virulent HPAIV in Calu3 is not enough. The replication kinetics lack positive controls.

RESPONSE: In response, we generated additional data in a modified Fig. 4F, adding virus replication kinetics in primary human airway cultures grown at air-liquid interfaces. Replication patterns were broadly similar to the Calu-3 cell line, and these results are discussed (lines 218-227). In this model, we compared late-timepoint replication kinetics to a representative seasonal rg-A/Texas/71/2017 (H3N2) virus as a fully adapted human virus control.

COMMENT #3. The experimental design (in chickens and ferrets) is extremely difficult to understand. The total number of animals in each experiment is not clear, particularly for ferrets (Fig 1B). The design should be clearly described. The legends should be clear enough to understand the figures. If possible, a diagram should be provided to explain the design of the experiments. Variation in the inoculation route of chickens Wigeon/SC/21 or Eagle/FL/22 should be justified. This can be the reason for variable pathogenesis of these two H5N1 viruses in chickens. Supplementary Fig S1, naïve birds were added 1 hpi or 1 dpi? 4 naïve chickens were added per donor chickens (legend of Fig S1) or per virus (L415 methods file)?

RESPONSE: We thank the reviewer for the recommendation to improve the clarity of Fig. 1 and Supplementary Fig. 1. We have updated both figures to include a schematic of the experimental design in panel A, and they include descriptions of the number of animals used, how they were housed, what type of samples were taken from the animals, and at what time points they were taken. Individual titers (symbols) have also been added to the bar graphs. We apologize for the error in the supplementary methods section where we say chickens were inoculated by different routes; both viruses were inoculated via the intranasal, intraocular, intraesophageal, intratracheal routes.

COMMENT #4. Reproducibility of the results is not rigorous enough. Triplicate wells, triplicate experiments ... duplicates... are vague descriptions. How many independent biological replicates have been done for each assay?

RESPONSE: We apologize for the lack of clarity which we have addressed in the revised

manuscript figures legends and/or materials and methods for individual assays. For example (Fig. 4C) now reads: “*Kinetics of pH inactivation of the Wigeon/SC/21, Eagle/FL/22, and CA/04 (H1N1)pdm09 viruses at 37°C. The data are shown as the mean ± SD from triplicate wells representing one of three independent experiments.*” (lines 252-254).

COMMENT #5. The interpretation of the results is very broad and should be rephrased and concentrated.

RESPONSE: Without reference to specific phrases, we were not entirely sure what interpretations the reviewer was suggesting we rephrase. We hope the additional material provided in response to reviewer 2’s comments go some way to addressing this concern, however.

Minor comments:

Main Text file:

COMMENT #1. Values in the figures (e.g. Figs 1B, 1C, 3B, 3C, supplementary Fig 2, supplementary Fig 7) should be provided as individual titres not as columns with mean+/-SD.

RESPONSE: As suggested, all figures containing virus titers from multiple animals have now been updated to reflect the individual values as well as the summary columns indicating mean values and SD.

COMMENT #2. L68-73, L410-421: The experimental design is very confusing. It is not clear how many birds were infected? whether all inoculated and sentinel chickens died or survived? What is the mean death time in chickens? Was there any difference in the pathogenicity in chickens (e.g. mean death time after inoculation with different route ...intranasal vs. intra esophageal?).“

RESPONSE: To clarify the experimental details, Supplementary Fig. 1 has been updated. We included the scheme of the experimental design indicating the number of animals used, what type of samples were taken from each animal, and when they were taken. Further, individual titers have been added to the summary data in the graphs to clarify the number of samples that were used to calculate the mean values. We have added the following information and presentation of the results at lines 71-78: “*Chickens (n = 3 per virus) directly inoculated with either Wigeon/SC/21 or Eagle/FL/22 died within 48 hours post-inoculation (hpi). Two of twelve contact chickens paired with Wigeon/SC/21 inoculated chickens shed virus and met euthanasia endpoints at 4 or 7 dpi, while two additional non-shedding contacts died at 4 or 10 dpi. Three of twelve contact chickens paired with Eagle/FL/22 inoculated chickens shed virus, and met euthanasia endpoints 3- 4 days post-inoculation (dpi); virus was transiently detected in a fourth bird which survived (Fig. 1B, and data not shown). There was no transmission detected from infected chickens to ferrets*”

COMMENT #3. L80: “Fig 1B” why 5 to 9 animals are shown?

RESPONSE: Fig. 1 has been updated to include the scheme of the experimental design and to clarify the protocol and number of animals used, along with when/what types of samples were taken from the animals. Fig. 1B (now Fig. 1C due to the addition of the experimental scheme) has also been updated to indicate individual titers instead of listing the number of animals above for enhanced clarity.

COMMENT #4. L85: Wigeon/SC/21 (4.3 log₁₀TCID₅₀/mL) is not reflected in Fig 1B

RESPONSE: Fig. 1B (now Fig. 1C) has been updated with individual titers such that the Wigeon/SC/21 value is now shown at 3 dpi.

COMMENT #5. L88: “TCID₅₀/mL” is not accurate... /gm, please check the legends of other (suppl.) figures.

RESPONSE: We have updated the manuscript to reflect the correct units (TCID₅₀/g [gram of wet tissues]), as well as in the figure legends (Fig. 1 and 3) and in Supplementary Fig. 7.

COMMENT #6. L119: ref.#8 is not suitable for this sentence.

RESPONSE: The original reference 8 referred to a database that contains a searchable tool with known markers. We agree that this is not particularly clear and have removed this reference and replaced it with:

8. Suttie, A., Deng, Y-M., Greenhill, A.R., Dussart, P., Horwood, P.F., Karlsson, E.A. Inventory of molecular markers affecting biological characteristics of avian influenza A viruses. *Virus Genes*. 55(6),739-768 (2019). PMID: 31428925

9. Lloren, K.K.S, Lee, T., Kwon, J.J., Song, M-S. Molecular markers for interspecies transmission of avian influenza viruses in mammalian hosts. *Int. J. Mol. Sci.* 18(12), 2706 (2017). PMID: 29236050

COMMENT #7. L121: Fig 2: for readability, please indicate the viruses used in the subsequent animal experiments by asterisks or arrows.

RESPONSE: The viruses used in subsequent animal experiments have been annotated by an asterisk and the figure legend for Fig. 2 has been updated accordingly.

COMMENT #8. L146-147: this sentence is not accurate. Fig. 3C shows low CK-NL-21 titres in the brain and intestine.

RESPONSE: The sentence has been corrected to reflect low or at the limit of detection Ck/NL/21 virus titers from the organs outside the respiratory tract.

COMMENT #9. L188: “the pH of HA activation” why the difference of 6 hpi and 24 hpi for the H5N1 and A(H1N1)pdm09 virus? Please explain!

RESPONSE: This assay measures the properties of HA on the surface of infected cells and, thus, is dependent on viability of the cells and replication kinetics of the viruses. We initially looked at 24 hpi for both viruses, but the A(H5N1) infected cells had undergone substantial CPE and syncytia formation could not be measured. We reverted to a published time point (6 hpi) for the H5N1 influenza virus (Reed et al. *J. Virol.* 84(3), 1527–1535, 2010). It is worth noting here that trypsin was not added to these cultures, so only single rounds of replication of the A(H1N1)pdm09 virus occurred. Limiting the A(H5N1) virus to 6 h also limited it to one replication cycle.

COMMENT #10. L189: ref.#11 is not suitable for this sentence

RESPONSE: Reference 11 discusses HA mediated immunity and how this can modulate zoonotic risk. In response to the concern, we have added another reference (now reference #12) that addresses NA immunity: 12. Sandbulte, M.R., Jimenez, G.S., Boon, A.C., Smith, L.R., Treanor, J.J., Webby, R.J. Cross-reactive neuraminidase antibodies afford partial protection

against H5N1 in mice and are present in unexposed humans. PLoS Med. 4(2), e59. 2007. PMID: 17298168

Supplementary file:

COMMENT #11. A list of viruses (not only the H5N1 mentioned in Table S1) used in this study and their accession numbers should be provided as a supplementary Table.

RESPONSE: Per the reviewer's suggestion, we generated additional Supplementary Tables 3 and 4 and provided the list of viruses used in the study and their GenBank or GISAID accession numbers, respectively.

COMMENT #12. Supplementary Figure S1: the figure and legend are confusing. L34, which values? The numbers refer to the contact animals (4 not 12?) excreted viruses, or seropositive? What about numbers for primarily inoculated animals? For clarity and transparency, the individual titres should be shown, not only the average +/- SD. "intranasally by natural routes" should be clarified.

RESPONSE: Supplementary Fig.1 has been updated with a new scheme of the experimental design (Supplementary Fig. 1A) to clarify the protocol and number of animals. We have also added individual titer data from chicken cloacal swabs (Supplementary Fig. 1B) in addition to the summary columns and added ferret data (Supplementary Fig. 1C) to complete the information gathered from the experimental design. The figure legend has been updated to clarify the routes of infection.

COMMENT #13. Supplementary Fig S6: the color coding is very difficult to follow particularly in panels E and F. It is recommended to use open and closed circles, triangles, ... for clarity.

RESPONSE: Supplementary Fig. 6 has been updated with a new color scheme and different symbols for clarity.

COMMENT #14. Supplementary L141: typo, the title is not complete. The HI test is not described in the materials and methods section. How was the test done and what about the SD?

RESPONSE: We corrected the title in Supplementary Table 2 and provided description of the hemagglutination inhibition (HAI) assay in Supplementary Materials and Methods section of the revised manuscript (lines 444-452). The results are the arithmetic mean titer of positive sera (HI titer >10).

COMMENT #15. Supplementary Table 3. Please describe how the Relative inactivity index was calculated and whether this index is statistically significance or not. Seroconversion data should be added to this table.

RESPONSE: We added a description of the scoring system used for assessment of the activity levels of ferrets inoculated with influenza A(H5N1) virus to the footnote of Supplementary Table 3 (now Supplementary Table 5). Because relative inactivity index is based on the subjective evaluations of ferret's activity, statistical significance of the data was not determined. We added a column entitled "Range of post infection HI titers" to the Table and provided seroconversion data.

COMMENT #16. Supplementary L205-213: the ATCC numbers should be added.

RESPONSE: We provided the ATCC catalog numbers for the cell lines used in the study in Supplementary Materials and Methods section of the revised manuscript.

COMMENT #17. Supplementary L225: MOI 0.01 or MOI 0.001? Please check the legend of figure 4.

RESPONSE: We corrected the MOI in Supplementary Materials and Methods section of the revised manuscript. An MOI of 0.001 was used for assessment of viral growth kinetics in Calu-3 cells (Fig. 4E).

COMMENT #18. Supplementary L230-231: The results for the two replicates should be shown.

RESPONSE: The data from two independent experiments were similar and trends were consistent between the two; we've chosen to represent only one of each in Fig. 4 for simplicity. Each experiment and data point in the replication kinetics experiments includes triplicate data points, per virus, per timepoint.

COMMENT #19. L391: where is the result of the “small intestine” for the initial animal experiment?

RESPONSE: Influenza Wigeon/SC/21 and Eagle/FL/22 virus replication was not examined in the samples from small intestine of ferrets. We revised the statement to read: “... *tissue samples were collected from the respiratory tract (nasal cavity, trachea, and lungs), the central nervous system (a combination of the brain stem, cerebellum, and frontal lobes), and the small intestine (not examined for Wigeon/SC/21 and Eagle/FL/22)*”. (lines 410-412).

COMMENT #20. L395: “every 2 days“ is not accurate (10 dpi)

RESPONSE: Per the reviewer's comment, we revised the statement to read: “*Ferrets were monitored daily, and nasal washes were collected at 1, 3, 5, 7 and 10 dpi, starting just before the inoculated and contact ferrets were co-housed*”.

COMMENT #21. Animal welfare issues, L418-419: please describe the housing and distance between chickens (prey) and ferrets (predator) (see PMID: 26202239) and describe the behaviour of animals in this experimental setting.

RESPONSE: The experimental design in Supplementary Fig. 1 has been updated to include a picture of the caging system for chicken-to-ferret transmission. The methods section has correspondingly been revised. We agree with the reviewer that it seems unlikely that co-housing a ‘predator’ species with a ‘prey’ species in separate cages in a confined space would be feasible. We spent more than two years testing this system with our Institutional Animal Care and Use Committee (IACUC), with extensive veterinary involvement, and with manipulation of distance and contact time. We never observed aggressive behavior from the ferrets, or defensive behavior from the chickens. We have hypothesized that this could be because both species of animal have been raised in a controlled laboratory environment with food and water ad-libitum. Neither species has ever been environmentally stressed, so this might explain why neither seem to mind that the other is nearby. We have worked with this system for several years, and we are confident in well-being of the animals.

Reviewer #3 (Remarks to the Author):

Major Comments:

None.

Minor Comments:

COMMENT #1. This reviewer would encourage the authors to move some of the figures/tables in the Supplemental Materials to the main manuscript. Several figures and tables in the Supplemental Materials are referenced extensively. It would enhance interpretation of the findings if some of this data was in the main manuscript.

RESPONSE: We thank the reviewer for this suggestion and have increased the volume of data representation in the revised manuscript by including the scheme of the experimental design (Fig 1A) and addition of a figure to panels of Fig. 4. We do, however, concede that we have contained a lot of valuable primary data in the Supplementary material. According to the rules for the authors, a limit of 10 displayed objects (Tables and /or Figures) exists, limiting how many Tables and/or Figures can be moved to the main manuscript. Currently we have not moved additional figures/tables to the main text but are very prepared to do so at the suggestion/approval of the Editor.

COMMENT #2. Lines 74-76. The authors indicate that ferret-ferret and chicken-ferret transmission studies have been performed; however, the data is not present in the manuscript. This text should either be clarified or removed, or the data should be added to the manuscript.

RESPONSE: We have revised the manuscript to clarify the animal experiments performed (See Fig. 1, and Supplementary Fig. 1), and this is now accurately reflected in the manuscript text. (lines 64-80).

COMMENT #3. Lines 143-144. The manuscript uses several isolates often with similar names. In this section, it would enhance clarity and ease of interpretation, if the peak viral titers for the isolates were presented in the same order as in Figure 3B and C.

RESPONSE: Per the reviewer's comment, we revised the statement to read: "*This trend extended to viral replication in tissues. Peak mean nasal wash titers were 2.9, 3.8, 5.4, and 6.0 log₁₀ TCID₅₀/mL for Ck/NL/21, Eagle/NC/22, Hawk/NC/22, and Scaup/GA/22 viruses, respectively*". (lines 165-167).

COMMENT #4. Line 192. This line appears to indicate that all 6 isolates in the paper were evaluated for receptor-binding preference. However, only 2 H5N1 isolates were used to assess receptor-binding preference. Please revise to clarify.

RESPONSE: We studied receptor-binding preference for Wigeon/SC/21 and Eagle/FL/22 viruses (Fig. 4, A and B). To clarify our findings, we deleted the sentence "*The relative strength of binding was similar among all viruses examined*".

COMMENT #5. The authors have not mentioned if there are any mutations in the HA that are known to be associated with binding to 2, 6 linked sialic acids and/or that are known to modify the pH of fusion. This should be added to the manuscript.

RESPONSE: None of the viruses studied in this report contained mutations known to be associated with 2, 6 linked sialic acid binding or HA fusion differences. We have now added to lines 279-282, "*In addition to the representative viruses we assayed phenotypically, we did not identify any mutations previously associated with 6'SLN-linked sialic acid binding or alterations in pH of HA activation in the viruses we sequenced.*"

COMMENT #6. Lines 202-203. The authors indicate that the polymerase activity of the viruses were similar over all temperature ranges. This is true when comparing the isolates, but it appears that polymerase activity at 37 and 40°C are lower than at 34°C. This finding requires more explanation. This observation of reduced activity at higher temperatures is unexpected as birds have higher core body temperatures (38-40°C). Moreover, for most influenza viruses when polymerase assays are performed, the overall level of polymerase activity is usually higher at 37°C relative to 33°C. When mutations such as PB2 E627K are introduced into avian viruses, these viruses usually have elevated activity at 33°C relative to the precursor strains; however, this activity typically is equal or less than that at 37°C. Thus, the authors are encouraged to discuss and/or explain the significance of these findings.

RESPONSE: The reviewer points out the perplexing increase in polymerase activity at 34°C vs 37/39°C. We too were surprised by these findings. To address, we considered the possibility that viability of our 293T cell line was impacted by temperature. We measured viability (Promega, CellTiterGlo) of 293T cells ($\approx 3 \times 10^4$ cells/well of 96-well plate, n = 8 wells/temperature) after 24 hr incubation at 34, 37, and 39°C. 293T cell viability was 28-29% higher at 34°C than 37/39°C (please see the data below). This could explain the higher polymerase activity observed at 34°C, and it suggests that our minireplicon cannot distinguish between these temperatures using this cell line. Since there were no statistical differences when comparing the two viruses at each individual temperature, the overall interpretation is the same. However, in a modified Fig. 4D we have retained only the 37°C temperature data and revised the text to remove temperature variation claims/conclusions.

COMMENT #7. Figure 4. It would be very interesting to evaluate viral growth kinetics at 33 or 34°C. This is not required for publication, but would potentially complement the polymerase activity data.

RESPONSE: We agree that temperature-dependent growth kinetics are an interesting avenue to explore, and we are encouraged by the Reviewers suggestion to do so. Given the results from comment #6, and in efforts to bolster the growth kinetics data, the revised manuscript now

includes an additional cell type (primary human airway cultures grown at air-liquid interfaces), which may add more physiological relevance.

COMMENT #8. Line 230, please indicate “data not shown” when discussing neutralizing titers against HA.

RESPONSE: We added “data not shown” when discussing neutralizing titers against HA. It now reads: “*As expected, we found no evidence of neutralizing HA antibodies against any of the A(H5N1) viruses (data not shown)*”. (lines 231-232).

Reviewer comments, second round review

Reviewer #1 (Remarks to the Author):

The authors have taken substantial efforts to address comments of all reviewers and added further experimental data. The revised version has markedly improved in readability and in the strength of arguments and conclusions.

All comments of this reviewer have been adequately addressed.

The manuscript should be published as is.

Reviewer #2 (Remarks to the Author):

Thank you for addressing my comments.

Reviewer #3 (Remarks to the Author):

Kandeil et al., have submitted a revised version of their manuscript describing the pathogenesis and transmission of clade 2.3.4.4b H5N1 viruses that are spreading in wild-birds in North America.

Major Comments: None.

Minor Comments:

1) Lines 72-80. Supplemental Figure 1 associated with section of the text describe additional experiments in which ferrets were housed in an isolator with infected chickens. This experiment is not described in the text of the manuscript. As a result, line 78 is confusing when the authors indicate no transmission was observed from chickens to ferrets. Please add additional text describing the chicken to ferret transmission study. This will clarify this section of the manuscript and help readers understand Supplemental Figure 1.

2) Line 78 indicates Fig 1B, but the text describes the results of chicken studies. Please change to Supplemental Fig 1B.

RESPONSES TO REVIEWERS' COMMENTS

Nature Communications manuscript NCOMMS-22-42769-A

Rapid evolution of A(H5N1) influenza viruses after intercontinental spread to North America

We appreciate the positive comments and minor suggestions on this second round of review. We have addressed the critiques from reviewer #3 below, adding a more thorough description of the chicken-to-ferret experiment and updating to the correct figure citations. Additional minor changes include revisions to the figure legend titles, as well as updated reference citations to reflect the most recent reports in what is an ongoing event.

REVIEWER COMMENTS

Reviewer #3:

Major Comments:

None.

Minor Comments:

COMMENT #1: Lines 72-80. Supplemental Figure 1 associated with section of the text describe additional experiments in which ferrets were housed in an isolator with infected chickens. This experiment is not described in the text of the manuscript. As a result, line 78 is confusing when the authors indicate no transmission was observed from chickens to ferrets. Please add additional text describing the chicken to ferret transmission study. This will clarify this section of the manuscript and help readers understand Supplemental Figure 1.

RESPONSE: We regret the oversight and have now added new text to describe the chicken-to-ferret and ferret-to-ferret studies. We've revised the associated results in two key areas stating, "Naïve chickens were placed in the same cage as inoculated birds and ferrets were housed in a separate cage 12 inches away from infected chickens (Supplementary Fig 1A)." (Page 5, lines 71-73) and, "No aerosol transmission was detected from infected chickens to naïve ferrets (Supplementary Fig. 1C), suggesting that the risk of transmission at the avian-mammalian interface remains low. We next assessed ferret to ferret contact transmission by placing inoculated ferrets in the same cage as naïve ferrets. Neither virus transmitted from inoculated ferrets to naïve direct-contact ferrets (Fig. 1A, and data not shown)." (Page 5, Lines 81-85).

COMMENT #2: Line 78 indicates Fig 1B, but the text describes the results of chicken studies. Please change to Supplemental Fig 1B.

RESPONSE: We have now revised and corrected all figure citations in the revised text from comment #1 (lines 67-90).